# AED: ADAPTABLE ERROR DETECTION FOR FEW-SHOT IMITATION POLICY

## ABSTRACT

We study the behavior error detection of few-shot imitation (FSI) policies, which behave in novel (unseen) environments. FSI policies would provoke damage to surrounding people and objects when failing, restricting their contribution to real-world applications. We should have a robust system to notify operators when FSI policies are inconsistent with the intent of demonstrations. Thus, we formulate a novel problem: adaptable error detection (AED) for monitoring FSI policy behaviors. The problem involves the following three challenges: (1) detecting errors in novel environments, (2) no impulse signals when behavior errors occur, and (3) online detection lacking global temporal information. To tackle AED, we propose Pattern Observer (PrObe) to parse the discernable patterns in the policy feature representations of normal or error states. PrObe is then verified in our seven complex multi-stage FSI tasks. From the results, PrObe consistently surpasses strong baselines and demonstrates a robust capability to identify errors arising from a wide range of FSI policies. Finally, the visualizations of learned pattern representations support our claims and provide a better explainability of PrObe.

## 1 INTRODUCTION

Few-shot imitation (FSI) is a critical and practical framework for future human-robot collaboration. Specifically, a robot solves a set of everyday missions through learning from a few owner's demonstrations. By constructing a *demonstration-conditioned (DC) policy*, existing mainstream FSI methods (Duan et al., 2017; James et al., 2018; Bonardi et al., 2020; Dasari & Gupta, 2020; Dance et al., 2021; Yeh et al., 2022; Watahiki & Tsuruoka, 2022) have made a significant breakthrough in novel environments. A major barrier that still limits their ability to infiltrate our everyday lives is to detect errors in novel (unseen) environments. Unlike other few-shot visual perception tasks, policies in FSI that fail to perform tasks in real scenes will cause severe impairments of surrounding objects and humans, resulting in rarely conducting real-world experiments (or only simple tasks).

Therefore, we propose a challenging and crucial problem: *adaptable error detection (AED)*, which aims to monitor FSI policies and report their behavior errors. In this work, behavior errors are the states inconsistent with the demonstrations, and the policy must be timely terminated when they occur. We emphasize that *addressing AED is as critical as enhancing FSI policies' performance*.

As illustrated in Figure 1, our AED problem introduces three novel challenges not present in a related problem, video few-shot anomaly detection (vFSAD). Specifically, (1) AED monitors the policy's behavior in novel environments whose normal states are not seen during training. Although vFSAD also works in an unseen scenario, the anomalies they monitor are common objects. (2) In AED, no impulse signals (i.e., notable changes) appear when behavior errors occur. Either (2-a) minor visual difference or (2-b) task misunderstanding is hard to recognize through adjacent rollout frames. In contrast, the anomaly in vFSAD raises a notable change in the perception field (e.g., a cyclist suddenly appearing on the sidewalk (Mahadevan et al., 2010)). (3) AED requires online detection to terminate the policy timely, lacking the global temporal information of the rollout. However, in vFSAD (Sultani et al., 2018; Feng et al., 2021), the whole video is accessible so that its statistical information can be leveraged, which is straightforward for detecting anomalies.

Unfortunately, previous methods cannot address all three challenges posed by our AED problem. First, one-class classification (OCC) methods, such as one-class SVMs (Zhou & Paffenroth, 2017) or autoencoders (Ruff et al., 2018; Park et al., 2018; Chen et al., 2020; Wong et al., 2021), struggle

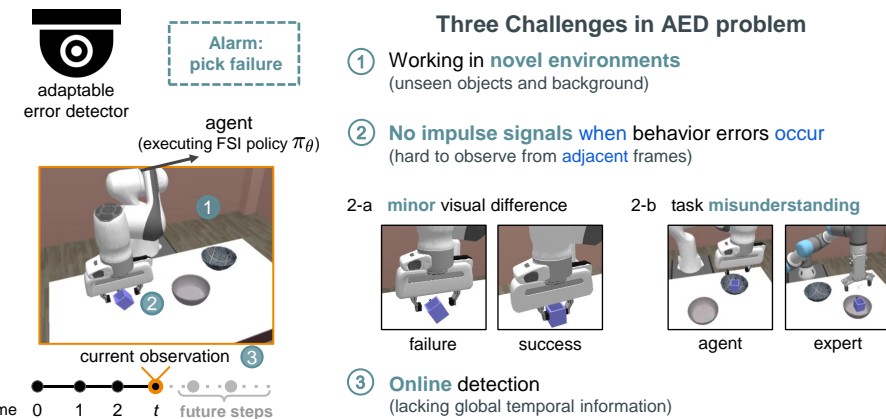

Figure 1: Challenges in adaptable error detection (AED) problem. To monitor the trained few-shot imitation (FSI) policy $\pi_\theta$, the adaptable error detector needs to address three challenges: (1) it works in novel environments, (2) no impulse signals (notable changes) when behavior errors occur, and (3) it requires online detection. These challenges make existing error detection methods infeasible.

due to their inability to handle unseen environments. Clearly, they are trained with normal-only samples and detect relying on a substantial error when encountering anomalies (errors), i.e., out-of-distribution (OOD) samples. However, normal states in novel environments containing unseen backgrounds and objects are already OOD to these methods, leading to poor performance. Second, multiple instance learning (Maron & Lozano-Pérez, 1997) or patch-based methods (Sultani et al., 2018; Roth et al., 2022), good at discovering tiny defects, may alleviate the (2-a) case in the second challenge. Even so, the minor visual difference and target misunderstandings cannot be recognized in novel environments if no consistent comparison between the current rollout and demonstrations is made, even at the instance/patch level. At last, due to lacking the crucial component (global temporal information from a completed trajectory), existing vFSAD methods (Sultani et al., 2018; Feng et al., 2021) are unsuitable in AED, where only partial temporal information exists.

As analyzed above, conventional methods run into insurmountable obstacles. To address AED, we discover the potential to leverage policy understanding, i.e., *how policy extracts feature representations in different situations (normal and error)*. In addition to obtaining the information from visual inputs, learning in policy features provides additional knowledge related to FSI tasks. We hypothesize that the feature representations (embeddings) from trained policies have implicit discernible patterns within the frames of normal and error states. Hence, the Pattern Observer (PrObe) is then proposed. Precisely, we collect and augment successful and failed rollouts in the base (training) environments. Then, the trained policy computes feature embeddings of rollout frames. In order to mitigate the impact of shifting into novel environments (first challenge), PrObe uses an extractor to parse patterns in policy features to obtain additional task-related knowledge. Next, PrObe computes the pattern flow containing pattern changes over time, efficiently leveraging existing temporal information (third challenge). Finally, PrObe conducts the consistent comparison (second challenge) by a fusion of pattern flow and transformed task-embeddings and distinguishes *when* the feature embedding is extracted, i.e., the frame that policy encountered in successful or failed rollouts. With these designs, PrObe can effectively address the challenging AED problem.

We designed seven challenging multi-stage FSI tasks (329 base and 158 novel environments in total) to validate PrObe's effectiveness. From the results, PrObe achieves the best performance and outperforms strong baselines with a significant gap of up to 40%. Meanwhile, it is the most robust method for different FSI policies. Also, we provide detailed ablations and the timing accuracy result to support our claims. Finally, a pilot study on addressing error correction after AED methods detect an error is conducted to demonstrate the practical scenario of the proposed problem.

**Contributions** (1) We formulate the adaptable error detection (AED) problem, accelerating the development of human-robot collaboration. (2) We propose PrObe, addressing AED by learning the pattern flow in the policy's feature representations. (3) We develop challenging FSI tasks to evaluate PrObe's effectiveness. It outperforms various baselines and is robust for distinct policies. Our work is expected to become an important cornerstone for FSI to conduct future real-world experiments.

## 2 RELATED WORKS

### 2.1 FEW-SHOT IMITATION (FSI)

**Policy**  With the recent progress in meta-learning (Finn et al., 2017a), the community explores the paradigm for learning policies from limited demonstrations during inference (Finn et al., 2017b; Yu et al., 2018; 2019a). Notably, these works either assume that the agent and expert have the same configuration or explicitly learn a motion mapping between them (Yu et al., 2018). Conversely, DC methods (Dance et al., 2021; Yeh et al., 2022; Watahiki & Tsuruoka, 2022) develop a policy that behaves conditioned both on the current state and demonstrations. Furthermore, they implicitly learn the mapping between agents and experts, making fine-tuning optional. Thereby, effectively extracting knowledge from demonstrations becomes the most critical matter. In most FSI works, no environment interactions are allowed before inference. Hence, policies are usually trained by behavior cloning (BC) objectives, i.e., learning the likelihood of expert actions by giving expert observations. Recently, DCRL (Dance et al., 2021) trains a model using reinforcement learning (RL) objects and performs FSI tasks without interacting with novel environments.

**Evaluation tasks**  Since humans can perform complex long-horizon tasks after watching a few demonstrations, FSI studies continuously pursue solving long-horizon tasks to verify if machines can achieve the same level. A set of research applies policy on a robot arm to perform daily life tasks in the real world (Yu et al., 2018) or simulation (Duan et al., 2017). The task is usually multi-stage and composed of primitives/skills/stages (Yeh et al., 2022; Hakhamaneshi et al., 2022), such as a typical pick-and-place task (Yu et al., 2019a) or multiple boxes stacking (Duan et al., 2017). Besides, MoMaRT (Wong et al., 2021) tackles a challenging mobile robot task in a kitchen scene.

Our work formulates the AED problem to monitor the FSI policies' behaviors and proposes PrObe to address AED, which is valuable to extensive FSI research. Besides, we also built seven challenging FSI tasks containing attributes like realistic simulation, task distractors, and various robot behaviors.

### 2.2 FEW-SHOT ANOMALY DETECTION (FSAD)

**Problem setting**  FSAD is an innovative research topic. Most existing studies deal with FSAD at the image level (Roth et al., 2022; Sheynin et al., 2021; Pang et al., 2021; Huang et al., 2022a; Wang et al., 2022) and only a few tackle video anomalies (Lu et al., 2020; Qiang et al., 2021; Huang et al., 2022b). Moreover, problem settings in the literature are diverse. Some works presume only normal data are given during training (Sheynin et al., 2021; Wang et al., 2022; Lu et al., 2020; Qiang et al., 2021), while others train models with normal and a few anomaly data and include unknown anomaly classes during performance evaluation (Pang et al., 2021; Huang et al., 2022b).

**Method summary**  Studies that only use normal training samples usually develop a reconstruction-based model with auxiliary objectives, such as meta-learning (Lu et al., 2020), optical flow (Qiang et al., 2021), and adversarial training (Sheynin et al., 2021; Qiang et al., 2021). Besides, patch-based methods (Roth et al., 2022; Pang et al., 2021) reveal the performance superiority on main image FSAD benchmark (Bergmann et al., 2019) since the anomaly are tiny defeats. Regarding video anomaly detection, existing works access a complete video to compute the temporal information for determining if it contains anomalies. In addition, vFSAD benchmarks (Mahadevan et al., 2010; Lu et al., 2013) provide frame-level labels to evaluate the accuracy of locating anomalies in videos.

In our work, we formulate the AED problem. As stated earlier, addressing AED is more challenging than vFSAD since it brings two more challenges, including no impulse signals and online detection. Moreover, previous error detection methods are infeasible for the proposed AED problem.

## 3 PRELIMINARIES

**Few-shot imitation (FSI)**  FSI is a framework worthy of attention, accelerating the development of human-robot collaboration, such as robots completing pre-tasks alone or interacting with humans for highly complex tasks. Following Yeh et al. (2022), a FSI task is associated with a set of base environments $E^b$ and novel environments $E^n$. In each novel environment $e^n \in E^n$, a few demonstrations $\mathcal{D}^n$ are given. The objective is to seek a policy $\pi$ that achieves the best performance

(e.g., success rate) in novel environments leveraging few demonstrations. Notably, the task in base and novel environments are semantically similar, but their backgrounds and interactive objects are disjoint. Besides, observations and demonstrations are RGB-D images that only provide partial information in recent studies (Dasari & Gupta, 2020; Yeh et al., 2022).

Additionally, there are several challenges when addressing FSI tasks in practice, including (1) the task is a multi-stage task, (2) demonstrations are length-variant and misaligned, and (3) the expert and agent have a distinct appearance or configuration, as mentioned in Yeh et al. (2022). Developing a policy to solve the FSI problem and simultaneously tackle these challenges is crucial.

**Demonstration-conditioned (DC) policy**    As previously stated, the expert and agent usually have different appearances or configurations in practice. The DC policy $\pi(a \mid s, \mathcal{D})$ learns implicit mapping using current states $s$ and demonstrations $\mathcal{D}$ to compute agent actions $a$. Next, we present the unified architecture of DC policies and how they produce the action. When the observations and demonstrations are RGB-D images that only provide partial information, we assume that current history $h$ (observations from the beginning to the latest) and demonstrations are adopted as inputs.

A DC policy comprises a feature encoder, a task-embedding network, and an actor. After receiving the rollout history $h$, the feature encoder extracts the history features $f_h$. Meanwhile, the feature encoder also extracts the demonstration features. Then, the task-embedding network computes the task-embedding $f_\zeta$ to retrieve task guidance. Notably, the lengths of agent rollout and demonstrations can vary. The task-embedding network is expected to handle length-variant sequences by padding frames to a prefixed length or applying attention mechanisms. Afterward, the actor predicts the action for the latest observation, conditioned on the history features $f_h$ and task-embedding $f_\zeta$. Besides, an optional inverse dynamics module predict the action between consecutive observations to improve the policy's understanding of how actions affect environmental changes. At last, the predicted actions are optimized by the negative log-likelihood or regression objectives (MSE).

## 4    AED PROBLEM

We formulate AED to monitor FSI policies and report their behavior errors, i.e., states in policy rollouts inconsistent with demonstrations. The inevitable challenges faced in addressing AED is previously presented. Next, we formally state the problem.

**Problem statement**    Let $c$ denote the category of agent's behavior modes, where $c \in \mathbf{C}, \mathbf{C} = \{\text{normal}, \text{error}\}$. When the agent (executing trained $\pi_\theta$) performs FSI task in a novel environment $e^n$, an adaptable error detector $\phi$ can access the agent's rollout history $h$ and few expert demonstrations $\mathcal{D}^n$. Then, $\phi$ predicts the categorical probability $\hat{y}$ of the behavior mode for the latest state in $h$ by

$$\hat{y} = \phi(h, \mathcal{D}^n) = P(c \mid enc(h, \mathcal{D}^n)), \tag{1}$$

where $enc$ denotes the feature encoder, and it may be $enc_\phi$ ($\phi$ contains its own encoder) or $enc_{\pi_\theta}$ (based on policy's encoder). Next, let $y$ represent the ground truth probability, we evaluate $\phi$ via the expectation of detection accuracy over agent rollouts $X^n$ in all novel environments $E^n$:

$$\mathbb{E}_{e^n \sim E^n} \mathbb{E}_{e^n, \pi_\theta(\cdot \mid \cdot, \mathcal{D}^n)} \mathbb{E}_{h \sim X^n} A(\hat{y}, y), \tag{2}$$

where $A(\cdot, \cdot)$ is the accuracy function that returns 1 if $\hat{y}$ is consistent with $y$ and 0 otherwise. However, frame-level labels are lacking in novel environments since we only know if the rollout is successful or failed at the end. Hence, we present the details of $A(\cdot, \cdot)$ in the Appendix's Section E.1.

**Protocol**    To train the adaptable error detector $\phi$, we assume it can access the resources in base environments $E^b$ just like FSI policies, i.e., successful agent rollouts $X_{succ}^b$ and a few expert demonstrations $\mathcal{D}^b$, which are collected by executing the agent's and expert's optimal policies $\pi^{*1}$, respectively. Besides, we collect failed agent rollouts $X_{fail}^b$ by intervening in the agent's $\pi^*$ at critical timing (e.g., the timing to grasp objects) so that we have precise frame-level error labels.

---

[1]We expect the agent's optimal policy is known in base environments. Furthermore, successful agent rollouts are not demonstrations since the agent and expert have different configurations.

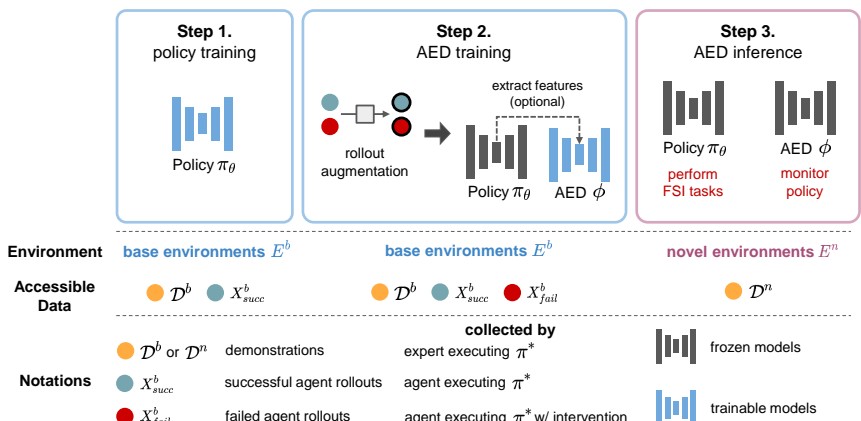

Figure 2: AED pipeline. Before training, we collect successful agent rollouts $X_{succ}^b$, failed agent rollouts $X_{fail}^b$, and a few expert demonstrations $\mathcal{D}^b$ for all base environments $E^b$. Then, the process contains three phases: policy training, AED training, and AED inference. The adaptable error detector $\phi$ aims to report policy $\pi_\theta$'s behavior errors when performing in novel environments $E^n$.

Figure 2 shows the process: (1) Policy $\pi_\theta$ is optimized using successful agent rollouts $X_{succ}^b$ and few demonstrations $\mathcal{D}^b$. (2) Adaptable error detector $\phi$ is trained on agent rollouts $X_{succ}^b$, $X_{fail}^b$ and few demonstrations $\mathcal{D}^b$. Besides, $\phi$ extracts features from policy $\pi_\theta$ if it depends on $\pi_\theta$'s encoder. And $\pi_\theta$'s parameters are not updated in this duration. (3) $\phi$ monitors policy $\pi_\theta$ performing FSI tasks in novel environments $E^n$. No agent rollouts are pre-collected in $E^n$; $\pi_\theta$ solves the task leveraging few demonstrations $\mathcal{D}^n$. AED aims to evaluate the detector $\phi$'s ability to report $\pi_\theta$'s behavior errors.

## 5    PATTERN OBSERVER (PROBE)

**Overview**    Our motivation is to detect behavior errors in the space of policy features to acquire additional task-related knowledge. Thus, we propose Pattern Observer (PrObe), an adaptable error detector, to discover the unexpressed patterns in the policy features extracted from frames of successful or failed states. Even if the visual inputs vary during inference, PrObe leverages the pattern flow and a following consistent comparison to alleviate the challenges in Figure 1. As a result, it can address the crucial AED problem effectively. Furthermore, a rollout augmentation is designed to increase the collected rollout diversity and enhance method robustness, as we now present in detail.

### 5.1    ROLLOUT AUGMENTATION

To obtain balanced (successful and failed) data and precise frame-level labels, we collect $X_{succ}^b$ and $X_{fail}^b$ using the agent's optimal policy $\pi^*$ with intervention rather than the trained policy $\pi_\theta$. However, even if $\pi_\theta$ is trained on the rollouts collected by the agent executing $\pi^*$, the trained $\pi_\theta$ will still deviate from agent's $\pi^*$ due to the limited rollout diversity, i.e., a behavior bias exists between the two policies, which should be reduced. Otherwise, an adaptable error detector trained on raw rollouts ($X_{succ}^b$ and $X_{fail}^b$) will yield a high false-positive during AED inference.

Accordingly, we augment agent rollouts so as to increase rollout diversity and dilute the behavior bias between trained $\pi_\theta$ and agent's $\pi^*$. Specifically, we iterate each frame (and its label) from sampled agent rollouts and randomly apply the following operations: *keep*, *drop*, *swap*, and *copy*, with probability 0.3, 0.3, 0.2, and 0.2, respectively. As a result, we inject distinct behaviors into collected rollouts, such as speeding up (interval frames dropped), slowing down (repetitive frames), and not smooth movements (swapped frames). We carefully verify its benefits in Section 6.

### 5.2    PROBE ARCHITECTURE

As depicted in Figure 3, PrObe comprises three major components: a pattern extractor, a pattern flow generator, and an embedding fusion. Since PrObe detects errors through policy features, the history features $f_h$ and task-embeddings $f_\zeta$ from the trained policy $\pi_\theta$ are passed into PrObe.

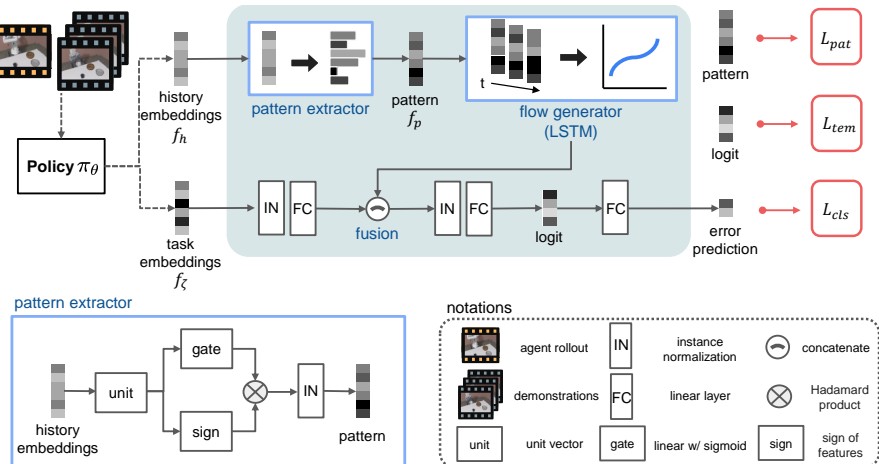

Figure 3: Architecture of PrObe. PrObe detects behavior errors through the pattern extracted from policy features. The learnable gated pattern extractor and flow generator (LSTM) compute the pattern flow of history features $f_h$. Then, the fusion with transformed task-embeddings $f_\zeta$ aims to compare the task consistency. PrObe predicts the behavior error based on the fused embeddings. Objectives, $L_{pat}$, $L_{tem}$, and $L_{cls}$, optimize the corresponding outputs.

At first, the pattern extractor aims to gather discriminative information from each embedding of history features $f_h$ (the features of a rollout sequence). In more detail, $f_h$ is transformed into unit embeddings divided by its L2 norm, which mitigates biases caused by changes in visual inputs. Then, a learnable gate composed of a linear layer with a sigmoid function determines the importance of each embedding cell. A Hadamard product between the sign of unit embeddings and the importance scores, followed by an instance normalization (IN), is applied to obtain the pattern features $f_p$ for each timestep. PrObe then feeds $f_p$ into a flow generator (a LSTM) to generate the pattern flow. Intuitively, the changes in the pattern flow of successful and failed rollouts will differ.

On the other hand, the task-embeddings $f_\zeta$ is transformed by an IN layer, a linear layer (FC), and a tanh function. This step tries to map the task-embeddings into a space similar to pattern flow. Then, a fusion process concatenates the pattern flow and transformed task-embeddings to compute the error predictions $\hat{y}$. We expect this fusion process to be able to compare the consistency between agent rollout and tasks and use this as a basis for whether behavior errors occur.

**Objectives** PrObe is optimized by one supervised and two unsupervised objectives. First, since accurate frame-level error labels $y$ are available during training, a binary cross-entropy (BCE) objective $L_{cls}$ is used to optimize the error prediction $\hat{y}$. Second, $L_{pat}$ is a L1 loss applied to the pattern features $f_p$. The $L_{pat}$ objective encourages the pattern extractor to generate sparse pattern features, which are easier to observe the changes. In addition, a contrastive objective $L_{tem}$ is added to logit embeddings (the embedding before the final outputs) to reinforce the difference between the normal and error states in failed rollouts $X_{fail}^b$, i.e., we sample anchors from $X_{fail}^b$, and the positive and negative samples will be logits of the same or opposed states from the same rollout.

However, the logits of adjacent frames will contain similar signals since they contain temporal information from the pattern flow, even if behavioral errors have occurred (cf. Appendix's Figure 7). Blindly forcing the logits of normal and error states to be far away from each other may mislead the model and have a negative impact. Hence, $L_{tem}$ is a novel temporal-aware triplet loss (Schroff et al., 2015). The temporal-aware margin $m_t$ in $L_{tem}$ will be enlarged or reduced considering the temporal distance of anchor, positive and negative samples. The $L_{tem}$ objective is calculated by:

$$L_{tem} = \frac{1}{N} \sum_{i=0}^{N} \max(\|\text{logit}_i^a - \text{logit}_i^p\|_2 - \|\text{logit}_i^a - \text{logit}_i^n\|_2 + m_t, 0), \qquad (3)$$

where $m_t = m * (1.0 - \alpha * (d_{ap} - d_{an}))$, $m$ is the margin in the original triplet loss, and $d_{ap}$, $d_{an}$ are the clipped temporal distances between the anchor and positive sample and the anchor and

negative sample, respectively. And $N$ is the number of sampled pairs. Ultimately, the total loss $L_{total}$ combines $L_{pat}$, $L_{tem}$, $L_{cls}$ objectives weighted by $\lambda_{pat}$, $\lambda_{tem}$, and $\lambda_{cls}$.

# 6 EXPERIMENTS

We design the experiments to verify the following questions: (1) Is detecting behavior errors from policy features beneficial to addressing the novel AED problem? (2) Can the proposed PrObe precisely recognize the behavior errors at the accurate timing? (3) Do the embeddings learned by PrObe have discernable patterns? (4) How practical is our proposed AED research?

## 6.1 EXPERIMENTAL SETTINGS

**Evaluation tasks**   Existing robotic manipulation benchmarks (Yu et al., 2019b; James et al., 2020; Fu et al., 2020; Mandlekar et al., 2021) do not suit our needs since we formulate the novel AED problem. We design seven challenging multi-stage FSI tasks (six indoor and one factory scene) that contain the challenges introduced in Section 3. Detailed task descriptions, schematic, the number of environments, and distractors are shown in the Appendix's Section D. We build the tasks on Coppeliasim software (Robotics) and use the Pyrep (James et al., 2019) toolkit as a coding interface.

**FSI policies**   To verify if the AED methods can handle various policies' behaviors, we select three representative DC policies to perform FSI tasks, which are implemented following the descriptions in Section 3 and use the same feature extractor and actor architecture. The main difference between them is how they extract task-embeddings from expert demonstrations. NaiveDC[2] (James et al., 2018) concatenates the first and last frames of demonstrations and averages their embeddings as task-embeddings; DCT [3] (Dance et al., 2021) performs cross-demonstration attention and fuses them at the time dimension; SCAN (Yeh et al., 2022) computes the tasks-embeddings using stage-conscious attention to locate the critical frames in demonstrations.

**Error detection baselines**   We compare our PrObe with four error detection methods that possess different characteristics. Unlike PrObe, all methods are policy-independent (i.e., containing their own encoders) and use the same encoder architecture. We briefly introduce them: (1) **SVDDED**, an one-class SVM modified from Ruff et al. (2018), is a single-frame DC method trained with only successful rollouts. (2) **TaskEmbED**, a single-frame DC method trained with successful and failed rollouts, uses the NaiveDC (James et al., 2018) policy's strategy to extract task-embeddings when predicting errors. (3) **LSTMED** is a LSTM model (Hochreiter & Schmidhuber, 1997) predicts errors without demonstration information. (4) **DCTED** leverages the DCT (Dance et al., 2021) policy's strategy to predict errors. The detailed version can be found in the Appendix's Section C.

**Metrics**   The area under the receiver operating characteristic (AUROC) and the area under the precision-recall curve (AUPRC), two conventional threshold-independent metrics in error (or anomaly) detection literature, are reported. We linearly select 5000 thresholds spaced from 0 to 1 (or outputs of SVDDED) to compute the two scores. The accuracy function (presented in the Appendix's Section E.1) determines the error prediction results (i.e., TP, TN, FP, FN) at the sequence-level so that we further conduct and illustrate the timing accuracy experiment in Figure 5.

## 6.2 EXPERIMENTAL RESULT ANALYSIS

**Overall performance**   Our main experiment aims to answer the first question, i.e., is detecting the policy's behavior errors from its feature embedding beneficial? We let NaiveDC, DCT, and SCAN policies perform in dozens of novel environments for each task and record the results (rollout observations and the success or failure of the rollout). Policies behave differently, making it difficult for AED methods to effectively detect behavioral errors in all policies. We then let AED methods infer the error predictions and report the two metrics. It is worth noting that in this experiment, AUPRC has more reference value. Because the results of policy rollout are often biased (e.g., many successes or failures), AUPRC is less affected by the imbalance of labeling.

---

[2]Original name is TaskEmb, we renamed it to avoid confusion with error detection baselines.
[3]Original name is DCRL, we denote it as DCT (transformer) since we do not use RL objectives.

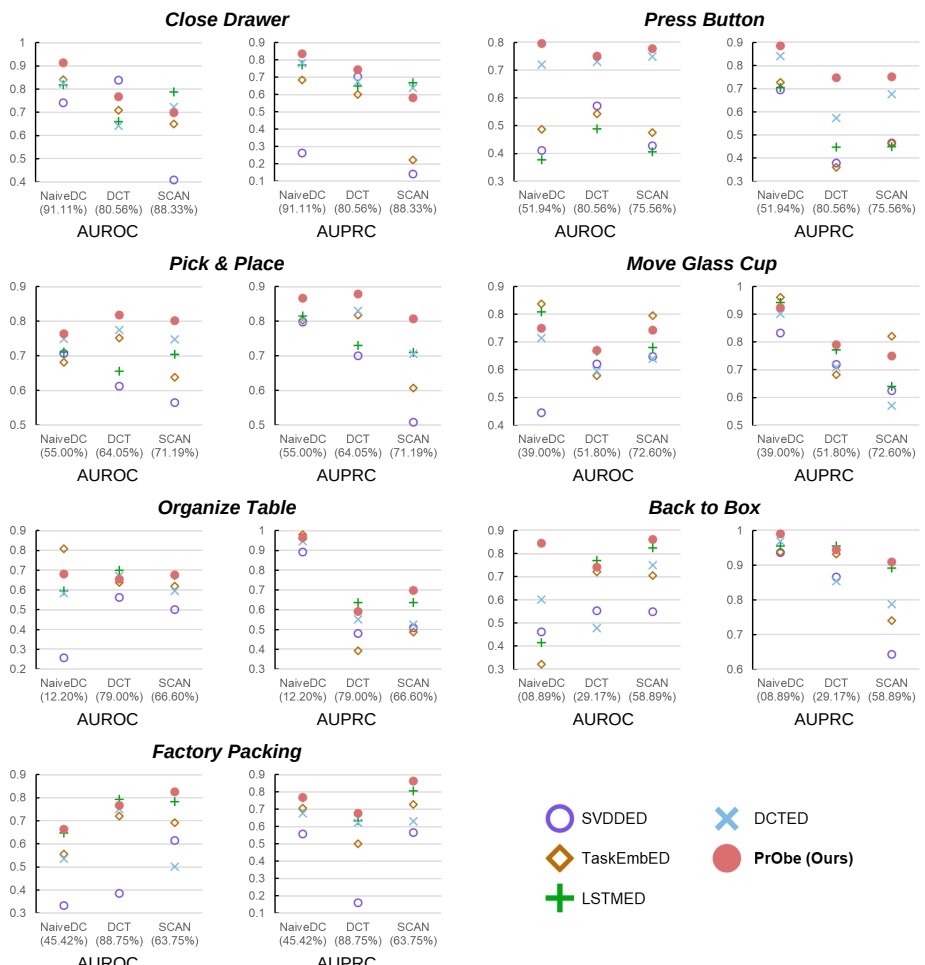

Figure 4: AED methods' performance on our seven challenging FSI tasks. The value under the policy indicates the policy's success rate in that task. The ranges of the two metrics [↑] are both 0 to 1. Our PrObe gets the best 13 AUROC and 15 AUPRC scores (out of 21 cases), demonstrating its superiority and robustness. We refer to the Appendix's Table 7 for detailed values.

From Figure 4, PrObe effectively reports the inconsistency between rollout frames and demonstrations (behavior errors) in novel environments, even some errors not included in the training data. Our PrObe obtained the best 13 AUROC and 15 AUPRC results out of 21 cases, which reveals its superiority among AED methods and robustness for different policies. Besides, we investigate the cases where PrObe got suboptimal performance and found that they can be grouped into two types: (1) Errors all occur at the same timing (e.g., at the end of the rollout) and can be identified without reference to the demonstrations (e.g., DCT policy in *Organize Table* task). (2) Errors are not obvious compared to the demonstrations, and there is still a chance to complete the task, such as the drawer being closed to a small gap (SCAN and DCT policies in *Close Drawer* task). In these cases, the pattern flow will not change enough to be recognized, resulting in suboptimal results for PrObe.

**Ablations** We conduct and summarize several ablations on the proposed components in the Appendix. First, Figure 11 indicates that the rollout augmentation (RA) strategy can increase the rollout diversity and benefits AED methods with temporal information. Second, Figure 12 proves that PrObe's design can effectively improve performance. Finally, Figure 13 shows PrObe's performance stability by executing multiple experiment runs on a subset of tasks and computing the performance variance. For an exhaustive version, please refer to the corresponding paragraphs in the Appendix.

**Timing accuracy** Our main experiment examines whether the AED methods can accurately report behavior errors when receiving the complete policy rollouts. To validate if they raise the error at the

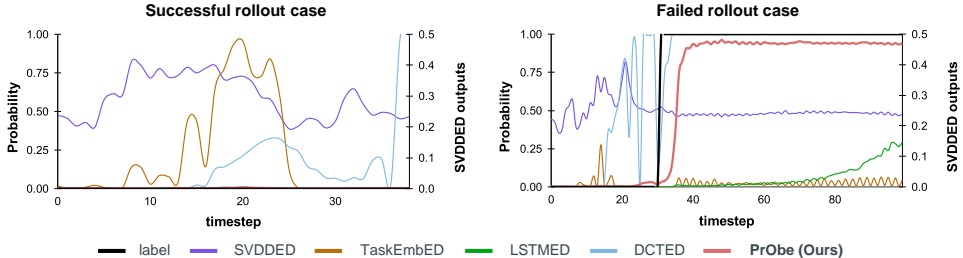

Figure 5: Visualization of timing accuracy. Raw probabilities and SVDDED outputs of selected successful rollout (left) and failed rollout (right) are drawn. Our PrObe can raise the error at the accurate timing in the failed rollout and stably recognizes normal states in the successful case.

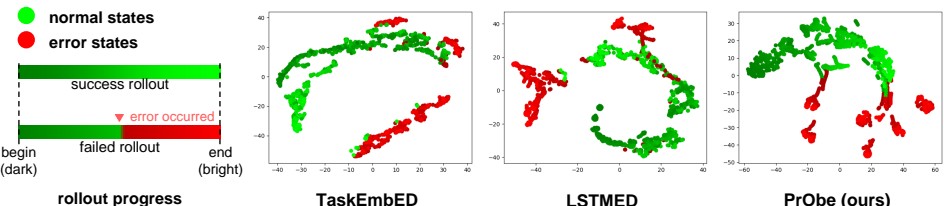

Figure 6: t-SNE visualization of learned embeddings (representations). The green and red circles represent normal and error states, respectively. The brightness of the circle indicates the rollout progress (from dark to bright). The embeddings learned by our PrObe have better interpretability because they exhibit task progress and homogeneous state clustering characteristics.

right timing, we manually annotate a subset of policy rollouts (SCAN policy in *Pick & Place* task) and visualize the raw outputs of all AED methods, where SVDDED outputs the embedding distance between observation and task-embedding while others compute the probabilities, as shown in Figure 5. In the successful rollout case, our PrObe and LSTMED stably recognize the states are normal, while other methods misidentify and increase the probability that inputs may be the error states. In the failed rollout case, PrObe detects the error after it has happened for a while ($< 5$ frames) and significantly increases the probability. We attribute this short delay to the fact that the pattern flow takes a short period to cause enough change. Even so, PrObe is the one that detects the error at the closest timing; other methods raise the probability too early or too late. We emphasize that this situation is ubiquitous and is not a deliberate choice.

**Embedding visualization**    To analyze if the learned embeddings possess discernable patterns, we present the t-SNE transform (van der Maaten & Hinton, 2008) of three AED methods' embeddings in Figure 6. The same 20 rollouts from the annotated dataset above are extracted by the three methods to obtain the embeddings. Obviously, the embeddings learned by TaskEmbED and LSTMED are scattered and have no explainable structure. In contrast, our PrObe's embeddings have task progress and homogeneous state clustering characteristics, i.e., the states with similar properties (e.g., the beginnings of rollouts, the same type of failures) are closer. This experiment supports the hypothesis that our PrObe can learn the implicit patterns from the policy's feature embeddings.

## 7 CONCLUSION

In this work, we point out the importance of monitoring policy behavior errors to accelerate the development of FSI research. To this end, we formulate the novel adaptable error detection (AED) problem, whose three challenges make previous error detection methods infeasible. To address AED, we propose the novel Pattern Observer (PrObe) by detecting errors in the space of policy features. With the extracted discernable patterns and additional task-related knowledge, PrObe effectively alleviates AED's challenges. It achieves the best performance, which can be verified by our main experiment and detailed ablations. We also demonstrate PrObe's superiority using the timing accuracy experiment and the learned embedding visualization. Finally, we provide the pilot study on error correction in the Appendix to confirm the practicality of our AED problem. Our work is an essential cornerstone in developing future FSI research to conduct complex real-world experiments.

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

## APPENDIX

This appendix presents more details of our work, following a similar flow to the main paper. First, Section A introduces the few-shot imitation (FSI) policies. Then, additional descriptions of PrObe's are provided in Section B. Besides, we introduce the implemented baselines of adaptable error detection in Section C. Next, in Section D, we exhaustively describe the designed multi-stage FSI tasks, including task configurations, design motivations, and possible behavior errors. In addition, more experimental results are shown in Section E to support our claims. At last, we responsibly present the limitations of this work in Section F.

## A FSI POLICIES DETAILS

We follow the main paper's Section 3 to implement three state-of-the-art demonstration-conditioned (DC) policies, including NaiveDC(James et al., 2018), DCT(Dance et al., 2021), and SCAN(Yeh et al., 2022), and monitor their behaviors.

**Architecture settings** In our work, we employ the same visual head and actor architecture for all policies to conduct a fair comparison. Table 1 and Table 2 present the settings of general components (visual head and actor) and respective task-embedding network settings for better reproducibility.

In the visual head, we added fully-connected (FC) and dropout layer after a single resnet18 to avoid overfitting the data from base environments. Furthermore, the input dimension of the actor module is dependent on the summation of the output dimensions from the visual head and the task-embedding network. For the DCT and SCAN policies, the task-embedding dimension matches the visual head's output. However, the NaiveDC policy has a task-embedding dimension that is twice the size.

In terms of the task-embedding networks, NaiveDC concatenates the first and last frames of each demonstration and computes the average as the task-embedding, without involving any attention mechanism. DCT, on the other hand, incorporates cross-demo attention to fuse demonstration features and then applies rollout-demo attention to compute the task-embedding. SCAN uses an attention mechanism to attend to each demonstration from the rollout, filtering out uninformative frames, and fuses them to obtain the final task-embedding. Besdies, We did not employ the bi-directional LSTM, which differs from the original setting in Yeh et al. (2022).

Table 1: General componets settings of FSI policies

| Visual head | |
| --- | --- |
| input shape | $T \times 4 \times 120 \times 160$ |
| architecture | a resnet18 with a FC layer and a dropout layer |
| out dim. of resnet18 | 512 |
| out dim. of visual head | 256 |
| dropout rate | 0.5 |
| Actor | |
| input dim. | 512 or 768 |
| architecture | a MLP with two parallel FC layers (pos & control) |
| hidden dims. of MLP | [512 or 768, 256, 64] |
| out dim. of pos layer | 3 |
| out dim. of control layer | 1 |

**Training details** To optimize the policies, we utilize a RMSProp optimizer (Hinton) with a learning rate of 1e-4 and a weight regularizer with a weight of 1e-2. Each training epoch involves iterating through all the base environments. Within each iteration, we sample five demonstrations and ten rollouts from the sampled base environment. The total number of epochs varies depending on the specific tasks and is specified in Table 4. All policy experiments, including both training and inference, are conducted on a Ubuntu 20.04 machine equipped with an Intel i9-9900K CPU, 64GB RAM, and a NVIDIA RTX 3090 24GB GPU.

Table 2: Task-embedding network settings of FSI policies

| NaiveDC | |
|---|---|
| input data | demonstrations |
| process | concat the first and last demonstration frames and average |
| out dim. | 512 |

| DCT (Transformer-based) | |
|---|---|
| input data | current rollout and demonstrations |
| process | a cross-demo attention followed by a rollout-demo attention |
| number of encoder layers | 2 |
| number of decoder layers | 2 |
| number of heads | 8 |
| normalization eps | 1e-5 |
| dropout rate | 0.1 |
| out dim. | 256 |

| SCAN | |
|---|---|
| input data | current rollout and demonstrations |
| process | rollout-demo attentions for each demonstration and then average |
| number of LSTM layer | 1 |
| bi-directional LSTM | False |
| out dim. | 256 |

# B  PROBE DETAILS

In this work, we propose the Pattern Observer (PrObe) to address the novel AED problem by detecting behavior errors from the policy's feature representations, which offers two advantages: (1) PrObe has a better understanding of the policy because the trained policy remains the same during both AED training and testing. This consistency enhances its effectiveness in identifying errors. (2) The additional task knowledge from the policy encoder aids in enhancing the error detection performance. We have described PrObe's components and their usages in the main paper. Here, we introduce more details about the design of model architecture and training objectives.

**Architecture design of PrObe**  The pattern extractor aims to transform the policy features into observable patterns. To achieve this, we can leverage discrete encoding, as used in Hafner et al. (2022). However, the required dimension of discrete embedding might vary and be sensitive to the evaluation tasks. To address this, we leverage an alternative approach that operates in a continuous but sparse space. Besides, for the pattern flow generator, we use a standard LSTM model instead of a modern transformer. This is motivated by the characteristics of the AED problem, where policy rollouts are collected sequentially, and adjacent frames contain crucial information when behavior errors occur. Thus, the LSTM model is better suited for addressing the AED problem. The contributions of PrObe's components are verified and shown in Figure 12. From the results, our designed components significantly improve the performance.

**Objectives**  We leverage three objectives to optimize our PrObe: a BCE loss $L_{cls}$ for error classification, a L1 loss $L_{pat}$ for pattern enhancement, and a novel temporal-aware triplet loss $L_{tem}$ for logit discernibility. First, the PrObe's error prediction $\hat{y}$ can be optimized by $L_{cls}$ since the ground-truth frame-level labels $y$ are accessible during training. The $L_{cls}$ is calculated by,

$$L_{cls} = -\frac{1}{N_r}\frac{1}{T_h}\sum_{i=0}^{N_r}\sum_{j=0}^{T_h}(y_{i,j}\cdot\ln\hat{y}_{i,j} + (1 - y_{i,j})\cdot\ln(1 - \hat{y}_{i,j})), \tag{4}$$

where $N_r$ and $T_h$ are the number and length of rollouts, respectively. Then, we leverage the L1 objective $L_{pat}$ to encourage the pattern extractor to learn a more sparse pattern embedding $f_p$, which can be formulated by

**Relationship between anchor and positive/negative sample**

Figure 7: Schematic of our novel temporal-aware triplet loss $L_{tem}$. When the anchor and positive sample are more related in both the time and task aspects, the temporal-aware margin $m_t$ will be larger (the closer the two embeddings are encouraged). Besides, $m_t$ provides a slight encouragement in case the positive sample and anchor are far away on temporal distance.

$$L_{pat} = \frac{1}{N_r}\frac{1}{T_h}\sum_{i=0}^{N_r}\sum_{j=0}^{T_h}|f_{p,i,j}|. \tag{5}$$

With the objective $L_{pat}$, the pattern embeddings are expected to be sparse and easily observable of changes, which benefits the model in distinguishing behavior errors. Next, as previously pointed out, when applying contrastive learning to temporal-sequence data, the time relationship should also be considered in addition to considering whether the frames are normal or errors for the task. Thus, $L_{tem}$ is a novel temporal-aware triplet loss (Schroff et al., 2015), and the temporal-aware margin $m_t$ in $L_{tem}$ will have different effects depending on the relationship between the anchor and the positive/negative sample, as depicted in Figure 7. The $L_{tem}$ can be calculated by:

$$L_{tem} = \frac{1}{N}\sum_{i=0}^{N}\max(\|\text{logit}_i^a - \text{logit}_i^p\|_2 - \|\text{logit}_i^a - \text{logit}_i^n\|_2 + m_t, 0), \tag{6}$$

where $m_t = m * (1.0 - \alpha * (d_{ap} - d_{an}))$, $m$ is the margin in the original triplet loss, and $d_{ap}$, $d_{an}$ are the clipped temporal distances between the anchor and positive sample and the anchor and negative sample, respectively. In addition, $N$ is the number of sampled pairs. With $L_{tem}$, PrObe can efficiently perform contrastive learning without getting confused by blindly pulling temporally distant anchors and positive samples closer. Lastly, the total loss $L_{total}$ is the combination of three objectives:

$$L_{total} = \lambda_{pat}L_{pat} + \lambda_{tem}L_{tem} + \lambda_{cls}L_{cls}, \tag{7}$$

where $\lambda_{pat}$, $\lambda_{tem}$, and $\lambda_{pat}$ are weights to balance different objectives.

## C   ERROR DETECTOR DETAILS

We compare our PrObe with several strong baselines that possesses different attributes, aiming to verify their effectiveness on addressing the adaptable error detection (AED) problem. In this section, we comprehensively present the attributes and training details associated with these baselines.

Table 3: Attributes of error detection methods

| method name | policy dependent | training rollouts | temporal information | DC-based | output type |
|---|---|---|---|---|---|
| SVDDED | | successful only | | ✓ | distance to center |
| TaskEmbED | | both | | ✓ | probability |
| LSTMED | | both | ✓ | | probability |
| DCTED | | both | ✓ | ✓ | probability |
| PrObe (ours) | ✓ | both | ✓ | ✓ | probability |

**Baseline attributes**    Table 3 presents the attributes of all error detection baselines (and our PrObe). All baselines have their own encoder and are independent of the policies, which offers the advantage that they only need to be trained once and can be used for various policies. However, as a result, they recognize errors only based on visual information. Now we describe the details of baselines:

- **SVDDED**: A modified one-class SVM method (Ruff et al., 2018) trained with only successful rollouts. It relies on measuring the distance between rollout embeddings to the center embedding to detect behavior errors, without considering the temporal information. We average demonstration features as the center and minimize the embedding distance between rollout features and the center during training.

- **TaskEmbED**: A single-frame baseline trained with successful and failed rollouts, which is a modification from the NaiveDC policy (James et al., 2018). It concatenates and averages the first and last demonstration frames as the task-embedding. Then, it predicts the behavior errors conditioned on the current frame and task-embedding.

- **LSTMED**: A standard LSTM model (Hochreiter & Schmidhuber, 1997) predicts errors based solely on the current rollout. It is trained with successful and failed rollouts with no demonstration knowledge provided. It is expected to excel in identifying errors that occur at similar timings to those observed during training. However, it may struggle to recognize errors that deviate from the demonstrations.

- **DCTED**: A baseline derived from the DCT policy (Dance et al., 2021) with temporal information. It is trained with successful and failed rollouts and incorporates with demonstration information. One notable distinction between DCTED and our PrObe lies in their policy dependencies. DCTED detects errors using its own encoder, which solely leverages visual information obtained within the novel environment. Conversely, PrObe learns within the policy feature space, incorporating additional task-related knowledge.

**Training details**    Similar to optimizing the policies, we utilize a RMSProp optimizer with a learning rate of 1e-4 and a weight regularizer with a weight of 1e-2 to train the error detection models. During each iteration within an epoch, we sample five demonstrations, ten successful agent rollouts, and ten failed rollouts from the sampled base environment for all error detection models except SVDDED. For SVDDED, we sample five demonstrations and twenty successful agent rollouts since it is trained solely on normal samples. Notably, all error detection experiments are conducted on the same machine as the policy experiments, ensuring consistency between the two sets of experiments.

## D    EVALUATION TASKS

To obtain accurate frame-level error labels in the base environments and to create a simulation environment that closely resembles real-world scenarios, we determined that existing robot manipulation benchmarks/tasks (Yeh et al., 2022; Yu et al., 2019b; James et al., 2020; Fu et al., 2020; Mandlekar et al., 2021) did not meet our requirements. Consequently, seven challenging FSI tasks containing 329 base and 158 novel environments are developed. Their general attributes are introduced in Section D.1 and Table 4; the detailed task descriptions and visualizations are provided in Section D.2 and Figure 8-9, respectively.

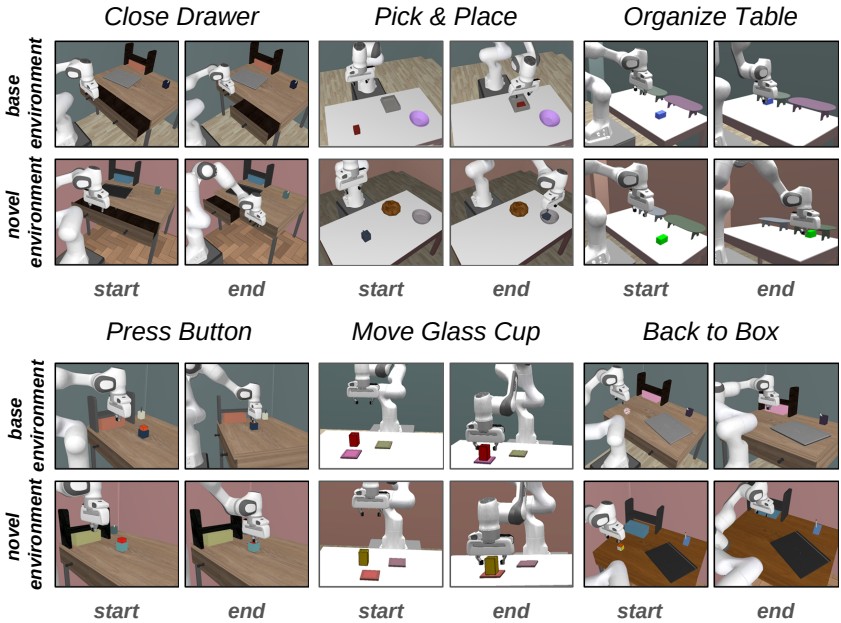

Figure 8: Visualizations of our designed indoor FSI tasks. The backgrounds (wall, floor) and interactive objects are disjoint between base and novel environments.

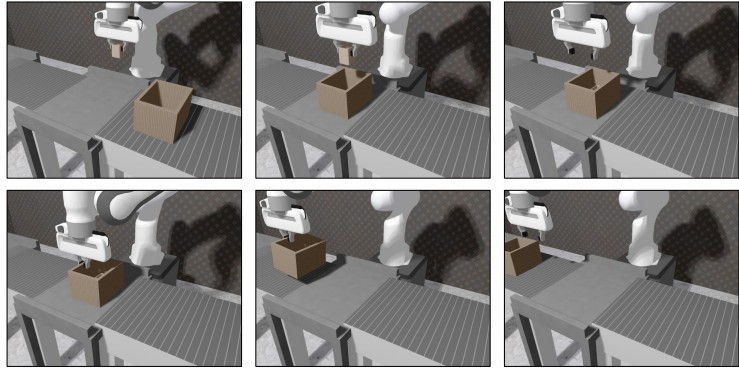

Figure 9: Progress visualization of our *Factory Packing* task. The agent needs to wait for the box's arrival. It then places the product into the box and lets the box move to the next conveyor belt.

## D.1 GENERAL TASK ATTRIBUTES

- **Generalization**: To present the characteristics of changeable visual signals in real scenes, we build dozens of novel environments for each task, comprising various targets and task-related or environmental distractors. Also, Dozens of base environments are built as training resources, enhancing AED methods' generalization ability. Letting them train in base environments with different domains can also be regarded as a domain randomization technique (Bonardi et al., 2020), preprocessing for future sim2real experiments.

- **Challenging**: Three challenges, including multi-stage tasks, misaligned demonstrations, and different appearances between expert and agent, are included in our designed FSI tasks. Moreover, unlike existing benchmarks (e.g., RLBench (James et al., 2020)) attaching the objects to the robot while grasping, we turn on the gravity and friction during the whole process of simulation, so grasped objects may drop due to unsmooth movement.

- **Realistic**: Our tasks support multi-source lighting, soft shadow, and complex object texture to make them closer to reality (see Figure 9). Existing benchmarks usually have no shadow (or only a hard shadow) and simple texture. The gravity and friction stated above also make our tasks more realistic.

## D.2 TASK DESCRIPTIONS

*Close Drawer (1-stage, indoor)*:

- **Description**: Fully close the drawer that was closed by the expert in demonstrations. Besides, the objects on the table will be randomly set in the designated area.
- **Success condition**: The target drawer is fully closed, and the drawer on the other side (distractor) must not be moved.
- **Behavior errors**: (1) Not fully close the target drawer. (2) Close the distractor, not the target drawer. (3) Close the two drawers at the same time.

*Press Button (1-stage, indoor)*:

- **Description**: Fully press the button that is randomly placed in a designated area.
- **Success condition**: The button is fully pressed.
- **Behavior errors**: (1) Not touching the button at all. (2) The button is not fully pressed.

*Pick & Place (2-stages, indoor)*:

- **Description**: Pick and place the mug/cup into the bowl the expert placed in demonstrations.
- **Success condition**: The mug/cup is fully placed in the target bowl.
- **Behavior errors**: (1) Failed to pick up the mug/cup. (2) Failed to place it into the target bowl. (3) Misplaced the mug/cup into the other bowl (distractor).

*Move Glass Cup (2-stages, indoor)*:

- **Description**: Pick up and place a glass of water on the coaster the expert placed in demonstrations.
- **Success condition**: The glass of water is placed on the target coaster and no water is spilled.
- **Behavior errors**: (1) Failed to pick up the glass of water. (2) drips spilling. (3) Not placed on the target coaster.

*Organize Table (3-stages, indoor)*:

- **Description**: Pick up and place the used object in front of the target bookshelf, and push it under the shelf.
- **Success condition**: The object is fully under the target shelf.
- **Behavior errors**: (1) Failed to pick up the object. (2) Losing object during moving. (3) Not fully placed under the target shelf.

*Back to Box (3-stages, indoor)*:

- **Description**: Pick up the magic cube/dice and place it into the storage box. Then, push the box until it is fully under the bookshelf.
- **Success condition**: The magic cube/dice is in the storage box, and the box is fully under the bookshelf.
- **Behavior errors**: (1) Failed to pick up the magic cube/dice. (2) Failed to place the magic cube/dice into the storage box. (3) The box is not fully under the bookshelf.

*Factory Packing (4-stages, factory)*:

- **Description**: Wait until the product box reaches the operating table, place the product in the box, and pick and place the box on the next conveyor belt.
- **Success condition**: The product is inside the box, and the box reaches the destination table.
- **Behavior errors**: (1) Failed to place the product into the box. (2) Failed to pick up the box. (3) Failed to place the box on the next conveyor belt.

Table 4: FSI task configurations

| Task | # of stages | # of epochs | # of base env. | # of novel env. | # of task distractors | # of env. distractors |
|------|------|------|------|------|------|------|
| *Close Drawer* | 1 | 160 | 18 | 18 | 1 | 3 |
| *Press Button* | 1 | 40 | 27 | 18 | 0 | 3 |
| *Pick & Place* | 2 | 50 | 90 | 42 | 1 | 0 |
| *Move Glass Cup* | 2 | 50 | 49 | 25 | 1 | 0 |
| *Organize Table* | 3 | 50 | 49 | 25 | 1 | 0 |
| *Back to Box* | 3 | 50 | 60 | 18 | 0 | 3 |
| *Factory Packing* | 4 | 80 | 36 | 12 | 0 | 0 |

# E  ADDITIONAL EXPERIMENTAL DETAILS

This section reports experimental details not presented in the main paper. We introduce the accuracy function used for determining an error prediction result in Section E.1. Besides, more experiment results are provided in Section E.2.

## E.1  ACCURACY FUNCTION FOR OUR AED PROBLEM

| rollout successful? | any error raised? | marked as |
|------|------|------|
| ✓ | ✓ | false positive (FP) |
| ✓ | ✗ | true negative (TN) |
| ✗ | ✓ | true positive (TP) |
| ✗ | ✗ | false negative (FN) |

Figure 10: Rules of the accuracy function $A(\cdot, \cdot)$ to determine the error prediction results.

As stated in the main paper, the frame-level label is lacking during inference because novel environments only indicate whether the current rollout is a success or failure. Hence, we follow similar rules in Wong et al. (2021) to define the accuracy function $A(\cdot, \cdot)$, i.e., how to determine the error prediction outcome, as elaborated in Figure 10. The $A(\cdot, \cdot)$ determines the prediction result at the sequence-level, which may not sufficiently reflect whether the timing to raise the error is accurate. Thus, we further conduct the timing accuracy experiment (shown in Figure 5 of the main paper).

## E.2  MORE EXPERIMENTAL RESULTS

**FSI policy performance**    The policy performance is originally reported in Figure 4 of the main paper. Here, we summarize the comprehensive overview in Table 5. It is important to note that the standard deviations (STD) are relatively large since they are calculated across novel environments rather than multiple experiment rounds. The difficulty variability of novel environments, such as variations in object size or challenging destination locations, leads to different performance outcomes for the policies. Besides, the policy will conduct 20 executions (rollouts) for each novel environment to get the success rate (SR). Then, the average success rate and its STD are calculated across those SRs. Furthermore, these rollouts are also used for later AED inference.

In general, NaiveDC policy achieves the best result in the simplest task because the task-embedding from the concatenation of the first and last two frames is enough to provide effective information. However, its performance will drop rapidly as the task difficulty increases. On the other hand, DCT policy excels in tasks where the misalignment between demonstrations is less severe since it fuses the task-embeddings at the temporal dimension. At last, SCAN policy outperforms on the more difficult tasks because its attention mechanism filters out uninformative demonstration frames.

Table 5: FSI policy performance. **Success rate (SR)** [↑] and standard deviation (STD) are reported. Notably, STD is calculated across **novel environments**, rather than multiple experiment rounds.

| | *Close Drawer* | *Press Button* | *Pick & Place* | *Organize Table* |
|---|---|---|---|---|
| NaiveDC | $\mathbf{91.11 \pm 20.85}\%$ | $51.94 \pm 18.79\%$ | $55.00 \pm 24.98\%$ | $12.20 \pm 13.93\%$ |
| DCT | $80.56 \pm 26.71\%$ | $\mathbf{80.56 \pm 11.65}\%$ | $64.05 \pm 19.77\%$ | $\mathbf{79.00 \pm 09.27}\%$ |
| SCAN | $88.33 \pm 13.94\%$ | $75.56 \pm 12.68\%$ | $\mathbf{71.19 \pm 15.38}\%$ | $66.60 \pm 23.18\%$ |

| | *Move Glass Cup* | *Back to Box* | *Factory Packing* |
|---|---|---|---|
| NaiveDC | $39.00 \pm 28.63\%$ | $08.89 \pm 09.21\%$ | $45.42 \pm 30.92\%$ |
| DCT | $51.80 \pm 23.99\%$ | $29.17 \pm 09.46\%$ | $\mathbf{88.75 \pm 07.11}\%$ |
| SCAN | $\mathbf{72.60 \pm 11.84}\%$ | $\mathbf{58.89 \pm 10.87}\%$ | $63.75 \pm 33.11\%$ |

Figure 11: Effectiveness of rollout augmentation (RA). **AUROC**[↑] and **AUPRC**[↑] are reported. Methods trained without RA (dark) and with RA(bright) and their performance gap are listed. RA is more beneficial for methods with time information (rightmost three columns). Also, the improvement from RA is more evident when the performance of FSI policy is higher (SCAN case).

**Main experiment's detailed values and ROC curves**   The main experiment result is originally reported in Figure 4 of the main paper. Here, Table 7 and Figure 15 present the main experiment's detailed values and ROC curves, which better demonstrate our PrObe's performance gap against other methods. To highlight again, our proposed PrObe obtains the best performance on both metrics and can handle the various behaviors from different policies.

**Rollout augmentation**   As stated before, the collected agent rollouts have limited rollout diversity. In this experiment, we aim to validate whether the rollout augmentation (RA) increases the rollout diversity so that error detectors can have better robustness. In Figure 11, we present the results of all error detection methods (trained w/ and w/o RA) monitoring policies solving *Pick & Place* task, with the following findings: (1) RA provides a minor or even negative influence on SVDDED, which can be expected. Since SVDDED is only trained on (single) frames from successful rollouts, the frame dropped by RA decreases the frame diversity. (2) RA benefits methods with temporal information (LSTMED, DCTED, and PrObe), especially when FSI policy performs better (SCAN case) because error detectors cannot casually raise an error, or they will easily make a false-positive prediction. The results indicate that methods trained with RA raise fewer false-positive predictions and demonstrate improved robustness to different policy behaviors.

**Contribution of PrObe components**   We verify the component contributions of the proposed PrObe, as shown in Figure 12, which results from monitoring the SCAN policy solving the *Pick & Plcae* task. The naive model (first column) removes the pattern extractor and uses a FC layer followed by an IN layer to transform history embeddings. Obviously, the designed components and objectives boost the performance, especially when adding the pattern extractor into the model. We attribute this to the extracted embedding (representation) patterns by our components are more informative and discernible.

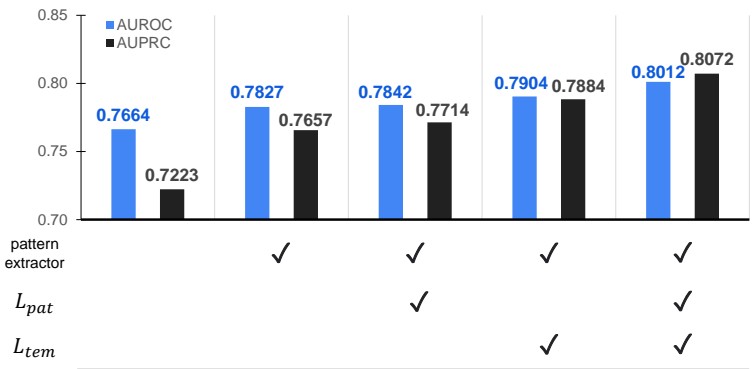

Figure 12: Component contributions. **AUROC**[↑] and **AUPRC**[↑] are reported. The pattern extractor and two auxiliary objectives significantly improve the performance.

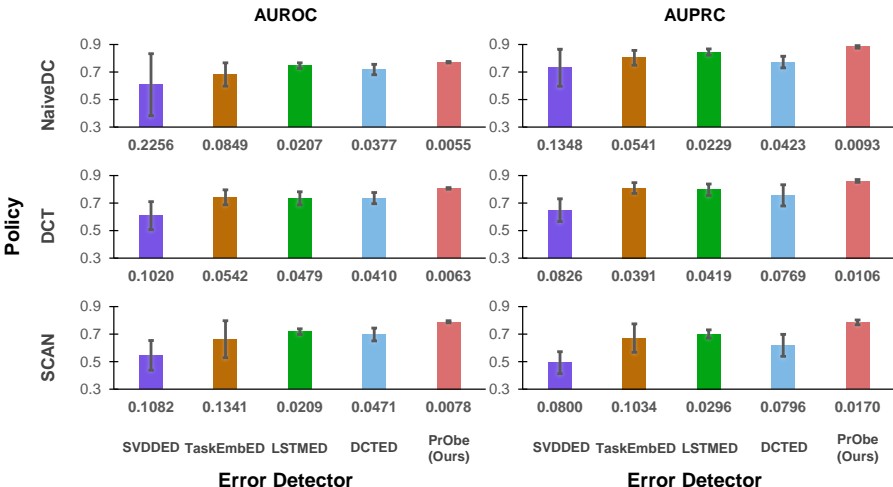

Figure 13: Performance results with error bars (standard deviation, STD). **AUROC**[↑], **AUPRC**[↑] are reported. Here, STDs are computed across multiple experiment rounds (random seeds). It is obvious that in the representative *Pick & Place* task, PrObe not only achieves the highest scores when detecting all policies' behavior errors but also has the best stability (the smallest STD).

**Performance stability** Our work involves training and testing of all policies and AED methods to produce the results for an FSI task. In addition, each task consists of numerous base and novel environments. These factors make it time-consuming and impractical to report the main experiment (Figure 4 of the main paper) with error bars. Based on our experience, generating the results of the main experiment once would require over 150 hours.

Therefore, we select the most representative *Pick & Place* task among our FSI tasks and conduct multiple training and inference iterations for all AED methods. We present the AUROC, AUPRC, and their standard deviations averaged over multiple experimental rounds (with five random seeds) in Figure 13. From the results, our proposed PrObe not only achieves the best performance in detecting behavior errors across all policies but also exhibits the most stable performance among all AED methods, with the smallest standard deviation.

**Case study on PrObe's failure prediction** We analyze the cases in which PrObe is poor at detecting behavior errors and visualize the results in Figure 14. As stated in the main paper, our PrObe requires a short period after the error occurs to make the pattern flow causing enough changes. However, in the Figure 14 case, the mug had fallen off the table, causing this rollout to be stopped

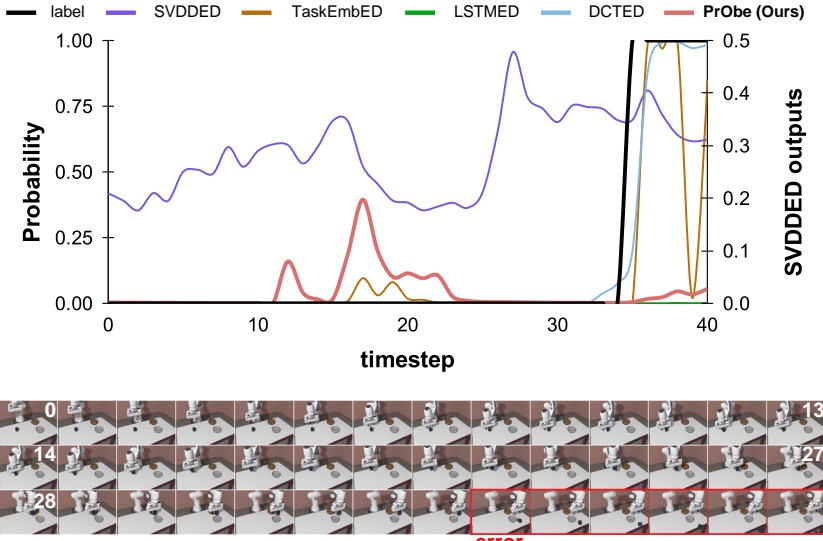

Figure 14: PrObe's failure prediction. In *Pick & Place* task, the rollout is terminated immediately if the mug is no longer on the table, which means the error detectors have only six frames to detect the error in this case. However, it took a while for our Probe's pattern flow to cause enough change. By the last frame, ProObe was just about to increase its predicted probability, but it was too late.

Table 6: Error Correction Results. **Success Rate (SR)** [↑] is reported. The detection threshold is set as 0.9 for all compared methods. The values indicate the performance of the DCT policy without correction (first column) and its collaboration with the correction policy and four AED methods (remaining columns). Our PrObe is the only method that can improve the performance since it detects the error most accurately.

| DCT policy | w/ TaskEmbED | w/ LSTMED | w/ DCTED | w/ PrObe |
|---|---|---|---|---|
| $80.56 \pm 11.65\%$ | $80.56 \pm 11.65\%$ | $80.28 \pm 11.60\%$ | $71.67 \pm 20.75\%$ | $\mathbf{82.22 \pm 10.17\%}$ |

immediately. Such time is insufficient for ProObe to reflect adequately, causing it to be unable to (substantially) increase the prediction probability of behavior errors at the last moment. Developing mechanisms for quick responsive pattern flow is one of our future research direction.

**Pilot study on error correction**  To further examine the practicality of our AED problem, we conducted a pilot study on error correction. In this experiment, the FSI policy is paused after the AED methods detect an error. Then, a correction policy from Wong et al. (2021), which resets the agent to a predefined safe pose, is applied. At last, the FSI policy continues to complete the task.

We conducted the study on *Press Button*, which will most likely be corrected after the error happens. In this task, the correction policy is defined as moving to the top of the table center. We let the DCT policy cooperate with the correction policy and four AED methods (SVDDED excluded), whose results are summarized in Table 6. We have two findings from the results: (1) Our PrObe is verified to be the most accurate method again. In contrast, other AED methods will raise the errors at the wrong timing, so the correction policy cannot benefit the performance (even negative influence for original success trials). (2) The performance gain from the correction policy is minor since it lacks precise guidance in unseen environments.

We believe error correction in novel environments deserves a separate study due to its challenging nature, as we observed in the pilot study. One potential solution is the human-in-the-loop correction works through instructions (Sharma et al., 2022; Cui et al., 2023) or physical guidance (Li et al., 2021). However, their generalization ability and cost when applying to our AED problem need further discussion and verification. We will leave it as our future work.

# F LIMITATIONS

**Offline evaluation?** Our ultimate goal is to terminate (or pause) the policy instantly when detecting a behavior error online. However, in our main experiment, only the trained policies performed in novel environments online, whose rollouts are collected. AED methods later use the offline rollouts to predict errors for a fair comparison. Nevertheless, we illustrate that PatObs can effectively address the online detection challenge through the time accuracy experiment. Also, the pilot study on error correction runs AED methods online to correct the error promptly.

**Not evaluated on benchmarks or in real-world environments?** Due to the practical challenges inherent in FSI tasks and the AED problem, none of the existing robot manipulation benchmarks align with our specific requirements. For instance, FSI tasks necessitate multi-stage scenarios or variations in configurations and appearances between experts and agents. In the case of AED, it is crucial to gather failed rollouts in the base environments efficiently. As a result, we have to develop our own environments and tasks.

Moreover, the lack of real-world experiments is caused by three factors: (1) Deploying the needed resources for the AED problem in the real world requires a high cost and long-term plan. (2) The FSI policies still fail to perform complex tasks in the real world, making assessments less than comprehensive. (3) Although our PrObe achieves the best performance, there is still significant room for improvement on the AED problem.

We emphasize that rashly conducting AED and FSI research in real environments may cause irreparable damage. Even previous error correction work (Wong et al., 2021) or related safe exploration research (Sootla et al., 2022; Anderson et al., 2023; Huang et al., 2023) were conducted in simulation environments, which shows that it is reasonable and valuable for us to conduct detailed verification in the simulation first. As mentioned earlier, we develop seven challenging and realistic multi-stage FSI tasks containing dozens of base and novel environments. To our knowledge, the simulation environments we established are the closest to the real scenarios in the literature.

Table 7: AED methods' performance on our seven challenging FSI tasks. The value under the policy indicates the policy's success rate in that task. Our proposed PrObe demonstrates its superiority among all the methods on both metrics and is the most robust method for different policies.

| Metric [↑] | AED | Close Drawer | | | Press Button | | |
| | | NaiveDC | DCT | SCAN | NaiveDC | DCT | SCAN |
| | | 91.11% | 80.56% | 88.33% | 51.94% | 80.56% | 75.56% |
| AUROC | SVDDED | 0.7404 | **0.8378** | 0.4079 | 0.4113 | 0.5710 | 0.4280 |
| | TaskEmbED | 0.8395 | 0.7081 | 0.6498 | 0.4872 | 0.5429 | 0.4754 |
| | LSTMED | 0.8186 | 0.6590 | **0.7867** | 0.3769 | 0.4886 | 0.4066 |
| | DCTED | 0.8250 | 0.6413 | 0.7218 | 0.7194 | 0.7306 | 0.7491 |
| | **PrObe** | **0.9133** | 0.7680 | 0.6978 | **0.7957** | **0.7506** | **0.7782** |
| AUPRC | SVDDED | 0.2626 | 0.7039 | 0.1411 | 0.6956 | 0.3789 | 0.4653 |
| | TaskEmbED | 0.6840 | 0.5995 | 0.2220 | 0.7280 | 0.3597 | 0.4657 |
| | LSTMED | 0.7700 | 0.6493 | **0.6677** | 0.7051 | 0.4468 | 0.4487 |
| | DCTED | 0.7813 | 0.6739 | 0.6390 | 0.8405 | 0.5734 | 0.6757 |
| | **PrObe** | **0.8350** | **0.7438** | 0.5829 | **0.8851** | **0.7474** | **0.7505** |

| Metric [↑] | AED | Pick & Place | | | Move Glass Cup | | |
| | | NaiveDC | DCT | SCAN | NaiveDC | DCT | SCAN |
| | | 55.00% | 64.05% | 71.19% | 39.00% | 51.80% | 72.60% |
| AUROC | SVDDED | 0.7074 | 0.6125 | 0.5646 | 0.4447 | 0.6201 | 0.6473 |
| | TaskEmbED | 0.6810 | 0.7523 | 0.6381 | **0.8370** | 0.5779 | **0.7950** |
| | LSTMED | 0.7116 | 0.6555 | 0.7050 | 0.8086 | 0.6670 | 0.6804 |
| | DCTED | 0.7493 | 0.7743 | 0.7482 | 0.7143 | 0.5954 | 0.6401 |
| | **PrObe** | **0.7635** | **0.8173** | **0.8012** | 0.7487 | **0.6697** | 0.7425 |
| AUPRC | SVDDED | 0.7971 | 0.7001 | 0.5078 | 0.8323 | 0.7188 | 0.6251 |
| | TaskEmbED | 0.8028 | 0.8173 | 0.6065 | **0.9609** | 0.6823 | **0.8204** |
| | LSTMED | 0.8148 | 0.7304 | 0.7103 | 0.9420 | 0.7720 | 0.6404 |
| | DCTED | 0.8041 | 0.8294 | 0.7063 | 0.9012 | 0.7086 | 0.5713 |
| | **PrObe** | **0.8665** | **0.8780** | **0.8072** | 0.9228 | **0.7907** | 0.7495 |

| Metric [↑] | AED | Organize Table | | | Back to Box | | |
| | | NaiveDC | DCT | SCAN | NaiveDC | DCT | SCAN |
| | | 12.20% | 79.00% | 66.60% | 08.89% | 29.17% | 58.89% |
| AUROC | SVDDED | 0.2581 | 0.5622 | 0.5000 | 0.4621 | 0.5537 | 0.5498 |
| | TaskEmbED | **0.8078** | 0.6382 | 0.6193 | 0.3221 | 0.7220 | 0.7041 |
| | LSTMED | 0.5946 | **0.7001** | 0.6734 | 0.4148 | **0.7707** | 0.8237 |
| | DCTED | 0.5844 | 0.6702 | 0.5962 | 0.6022 | 0.4772 | 0.7489 |
| | **PrObe** | 0.6808 | 0.6550 | **0.6759** | **0.8446** | 0.7276 | **0.8614** |
| AUPRC | SVDDED | 0.8911 | 0.4800 | 0.5086 | 0.9367 | 0.8652 | 0.6433 |
| | TaskEmbED | **0.9786** | 0.3922 | 0.4865 | 0.9375 | 0.9314 | 0.7401 |
| | LSTMED | 0.9611 | **0.6352** | 0.6356 | 0.9555 | **0.9546** | 0.8913 |
| | DCTED | 0.9452 | 0.5515 | 0.5255 | 0.9679 | 0.8534 | 0.7880 |
| | **PrObe** | 0.9652 | 0.5905 | **0.6983** | **0.9903** | 0.9438 | **0.9097** |

| Metric [↑] | AED | Factory Packing | | | Statistics | |
| | | NaiveDC | DCT | SCAN | Top 1 counts | Avg. Rank |
| | | 45.42% | 88.75% | 63.75% | | |
| AUROC | SVDDED | 0.3338 | 0.3849 | 0.6151 | 1 | 4.2 |
| | TaskEmbED | 0.5564 | 0.7201 | 0.6916 | 3 | 3.2 |
| | LSTMED | 0.6471 | **0.7934** | 0.7836 | 4 | 2.8 |
| | DCTED | 0.5361 | 0.7509 | 0.5006 | 0 | 3.2 |
| | **PrObe** | **0.6635** | 0.7670 | **0.8256** | **13** | **1.5** |
| AUPRC | SVDDED | 0.5583 | 0.1600 | 0.5657 | 0 | 4.5 |
| | TaskEmbED | 0.7057 | 0.5002 | 0.7287 | 3 | 3.5 |
| | LSTMED | 0.7667 | 0.6335 | 0.8063 | 3 | 2.5 |
| | DCTED | 0.6772 | 0.6224 | 0.6306 | 0 | 3.1 |
| | **PrObe** | **0.7676** | **0.6759** | **0.8622** | **15** | **1.4** |

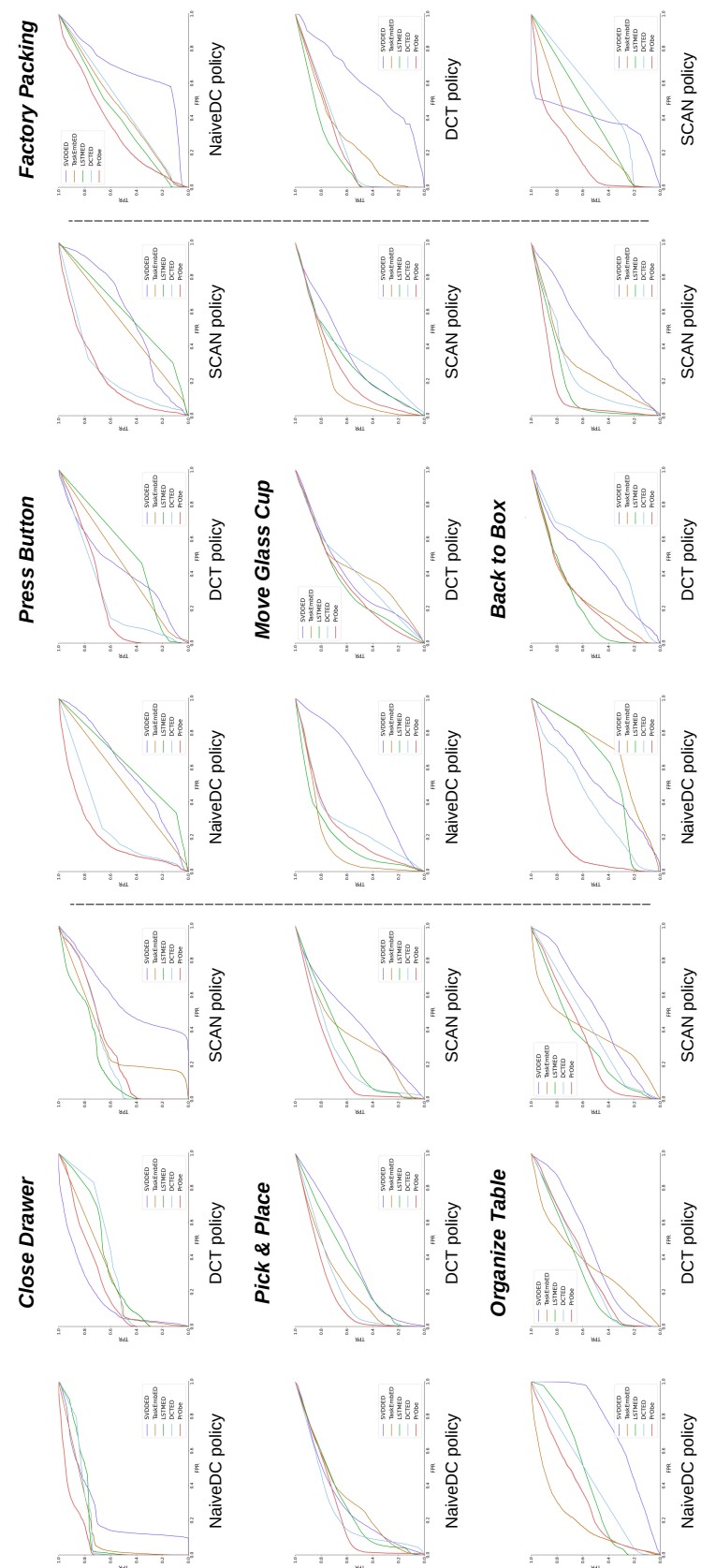

Figure 15: AED methods's ROC curves of all FSI tasks. The AUROC scores reported in Table 7 are the area under the curves.

