# OpenReview forum: "AED: Adaptable Error Detection for Few-shot Imitation Policy"
_ICLR.cc/2024/Conference — Submitted to ICLR 2024_

### Official Review · Reviewer_upjt · 2023-10-30

**Soundness:** 2 fair
**Presentation:** 3 good
**Contribution:** 3 good
**Rating:** 6
**Confidence:** 4

**Summary:**

This paper studies a novel problem: detecting behavior errors online in novel environments and name it adaptable error detection (AED). The problem setting comes with several challenges, including how to deal with unseen objects/backgrounds, subtle behavior differences between success and failure as well as detecting failure during policy execution.

The authors propose to learn pattern flow in the policy’s representation space and design novel architectures, such as a pattern extractor to extract discriminative patterns of different rollouts. It also applies several novel training tricks: rollout augmentation and temporal-aware triplet loss to stabilize training.

Besides the contribution of a new method, this paper also designs new challenging FSI tasks to evaluate AED methods and baselines.

**Strengths:**

- Studies an interesting problem that is essential for the deployment of robotic learning systems in safety-critical settings.
- The proposed method is novel and achieves superior performance compared to some error detection baselines.
- The designed tasks and paired environments (base and novel) will be beneficial for the community to study policy generalization and benchmark deployment performance.

**Weaknesses:**

- Baselines are not recent and strong enough for meaningful comparisons. All the baselines do not have access to the policy network while the proposed method utilizes it. There are a couple of recent works in error detection in robot manipulation, including MoMaRT [1], ThriftyDAgger [2], and some model-based approaches [3] worth comparing.

- Since both the policy and error detector are conditioned on demonstrations, it’s worth seeing the performance of the error detector under different demonstration qualities. When deploying a human-robot interactive system, we should be able to separate out human error and robot error.

- In the visualizations, the failure is detected after the fact (5 frames later), which is undesirable if the damage is already happened. This is expected as the design doesn’t include a world model to predict possible failures. I hope the authors can discuss this in more detail and point out the intended application of the proposed error detector.

[1] Wong et.al. Error-Aware Imitation Learning from Teleoperation Data for Mobile Manipulation

[2] Hoque et.al. ThriftyDAgger: Budget-Aware Novelty and Risk Gating for Interactive Imitation Learning

[3] Liu et.al. Model-Based Runtime Monitoring with Interactive Imitation Learning

**Questions:**

How does the method compare with recent papers, such as MoMaRT, ThrifyDAgger, etc.?

How does the PRAUC change with different demonstration qualities?

What’s the use case of the proposed system if it only detects failure after the fact?

---

> ### Author Response · Authors · 2023-11-11
> **Response to Reviewer upjt's review [1/2]**
>
> Dear Reviewer upjt,
>
> We sincerely appreciate your insightful comments and positive assessments. We now address the questions and issues you raised.
>
> *The issues listed in the weaknesses and questions are denoted as Wx and Qx, respectively, where x is the issue's index.*
>
> ---
>
>
> **What’s the use case of the proposed system if it only detects failure after the fact? (W3, Q3)**
>
> Thank you for this question.
>
> The fundamental assumption of our work is that  we are monitoring the few-shot imitation (FSI) policies, and they usually have the following settings:
>
> - **Training and testing environments are disjoint**. (base env. vs. novel env.)
> - **No exploration** before inference in novel environments.
> - **The expert only provides demonstrations** in the novel environment but may not be around the robot when it executes.
>
> These settings make our AED methods unable to get the information of human assistance (e.g., intervention, labeling new data) or world dynamic model as used in the references the reviewer listed. A practical scenario of our AED study is provided at the end of our response.
>
> Moreover, we are dealing with robot manipulation tasks, where the robot interacts closely with objects. Therefore, in our definition, an error (failure) occurs (right) after a failed interaction. From this moment, the current observation is inconsistent with demonstrations.
>
> In addition, to our knowledge, most existing studies related to failure detection on robot manipulation also detect the error after the fact, e.g., the reference [1] the reviewer listed.
>
> We are not saying that our setting is better than the one in [3], but the settings, targeted policies, and scenarios between ours and [3] are different. Thus, in our AED task, the goal is to detect the error as early as possible (after the fact). And our PrObe detects errors at the most accurate timing.
>
> Regarding the issue of delaying 5 frames, the camera collects the observations with fps=20. It means that our PrObe only causes a delay less than 0.25 seconds, which should be acceptable in practice. From our demo video, our PrObe can detect the error before the mug falls on the table. We believe the AED problem and proposed PrObe suit many real-world situations.
>
>
> **A practical scenario of our AED problem:**
>
> An intelligent robot (containing policy and AED method) is trained in the factory with daily life tasks. Then, it is sold to the customer's house to perform those tasks. The customer will provide few demonstrations for the robot but may not stay around the robot for the whole time. The robot itself should be able to recognize if it performs the task correctly or not. Also, it should terminate the policy and wait for expert's help or apply the recovery policy (in our appendix) when it detects a behavior mismatch. Note that in this case, the expert is the customer, who can provide demonstrations but may not be able to label corrected actions for the robot.
>
> ---
>
> **How does the method compare with recent papers, such as MoMaRT, ThrifyDAgger, etc.? (W1, Q1)**
>
> Thank you for recommending these references.
>
> As explained in Q3, in our setting, we don’t have access to human assistance (intervention and newly labeled data) and environmental exploration before inference, which are the natural limits of the FSI literature. Therefore, the works that need human-in-the-loop (reference [2]) or the world model (reference [3]) cannot be compared with our work.
>
> The reference [1] is already included in our references. However, as we pointed out in the Introduction, these reconstruction-based methods (one-class classification) suffer when dealing with our problem.
>
> They monitor the policies based on the *reconstruction errors of observations*. However, they will raise errors for **every frame in the novel environments** since they did not see them before. We have tried to add [1] as one of our baselines, but we cannot find appropriate thresholds since it yields large errors no matter if the frame is normal or an error in the novel environments.
>
> Moreover, we actually build our baselines **with various attributes**, including policy dependent, OCC (trained with only normal rollouts), temporal information, and demonstration-conditioned. Please find them in Appendix Table 3. Besides, there are also powerful models in our baselines. For example, DCTED (2021) is a transformer model, which should be more complicated than cVAE used in [1].
>
> However, we are grateful for receiving the references related to our work. We think summarizing and introducing these interactive error detection works is valuable, even though our settings are different. **We will include them in our final paper**.
>
> Please let us know if there is any other paper the reviewer thinks is related to us or fits our setting and should be compared.
>
> Thank you again for suggesting the references.
>
> ---
>
> Please find the rest of our response below.

---

> > ### Author Response · Authors · 2023-11-11
> > **Response to Reviewer upjt's review [2/2]**
> >
> > **How does the performance change with different demonstration qualities? (W2, Q2)**
> >
> > It’s really a good question.
> >
> > Indeed, the demonstration qualities might affect AED methods that leverage the demonstration features. We are now experimenting with this; please wait for our follow-up updates.
> >
> > Nevertheless, for now, we can still analyze the possible results from the method characteristics:
> >
> > 1. Those methods without accessing the demonstrations should not have significant performance differences, e.g., LSTMED.
> > 2. TaskEmbED should also obtain similar results if the task is still completed in the demonstration since it only uses the first and last frames of demonstrations.
> > 3. The remaining methods might be affected by the demonstration quality.
> >
> > Among them, SVDDED and DCTED should have a similar influence across different policies.
> >
> > However, our PrObe’s performance difference might depend on policies since it accesses the demonstration features from the policy.
> >
> > We will update the actual results once the experiment is completed. Thank you again for this interesting question.
> >
> > ---
> >
> > Please let us know if our responses have addressed your questions and issues.
> >
> > Besides, feel free to point out if there is any other issue or topic you are interested in or think will make our work more convincing.
> > We look forward to having active and insightful discussions with you!
> >
> > Thanks again for providing valuable comments and positive assessments.

---

> > > ### Author Response · Authors · 2023-11-13
> > > **Follow-up update on the demonstration quality experiment**
> > >
> > > Dear Reviewer upjt,
> > >
> > > We would like to share that the experiment on the demonstration qualities is completed. However, we found that observing the change patterns is a little hard since the *policy performance will also change* when adjusting the demonstration qualities.
> > >
> > > Specifically, the *policy performance usually drops* when the demonstration quality decreases (note that we are in the few-shot imitation problem). As a result, *the AED methods should get a higher AUPRC score* if they are not affected by the quality changes of demonstrations since it’s easier to find an error.
> > >
> > > ---
> > >
> > > Next, we present how the experiment is conducted. We test it on the *Press Button* task. We collect sub-optimal demonstrations that failed to press the button at the first trial but pressed the button successfully on the second try. We provide these sub-optimal demonstrations and let the policies and AED methods perform the task.
> > >
> > > Since our setting is few-shot, we have tested two cases, including
> > > 1. all five demonstrations are replaced with sub-optimal ones
> > > 2. only two out of five are sub-optimal demonstrations.
> > >
> > > The results are summarized below.
> > >
> > > ---
> > >
> > > - NaiveDC policy
> > >
> > > | Demonstration setting    |    Policy SR    | SVDDED | TaskEmbED | LSTMED |  DCTED |  PrObe |
> > > |--------------------------|:---------------:|:------:|:---------:|:------:|:------:|:------:|
> > > | All optimal              | 51.94% ± 18.79% | 0.6956 |   0.7280  | 0.7051 | 0.8405 | 0.8851 |
> > > | 3 optimal & 2 sub-optimal | 50.83% ± 18.50% | 0.6861 |   0.7407  | 0.7168 | 0.8832 | 0.9059 |
> > > | All sub-optimal | 51.67% ± 17.24% | 0.6414 |   0.7276  | 0.7044 | 0.8446 | 0.8930 |
> > >
> > > - DCT policy
> > >
> > > | Demonstration setting    |    Policy SR    | SVDDED | TaskEmbED | LSTMED |  DCTED |  PrObe |
> > > |--------------------------|:---------------:|:------:|:---------:|:------:|:------:|:------:|
> > > | All optimal              | 80.56% ± 11.65% | 0.3789 |   0.3597  | 0.4468 | 0.5734 | 0.7474 |
> > > | 3 optimal & 2 sub-optimal | 78.61% ± 14.02% | 0.3908 |   0.3811  | 0.4606 | 0.6019 | 0.7530 |
> > > | All sub-optimal | 77.22% ± 14.16% | 0.3899 |   0.4051  | 0.4718 | 0.5341 | 0.7588 |
> > >
> > > - SCAN policy
> > >
> > > | Demonstration setting    |    Policy SR    | SVDDED | TaskEmbED | LSTMED |  DCTED |  PrObe |
> > > |--------------------------|:---------------:|:------:|:---------:|:------:|:------:|:------:|
> > > | All optimal              | 75.56% ± 12.68% | 0.4653 |   0.4657  | 0.4487 | 0.6757 | 0.7505 |
> > > | 3 optimal & 2 sub-optimal | 71.67% ± 11.18% | 0.4985 |   0.5165  | 0.4979 | 0.7467 | 0.7804 |
> > > | All sub-optimal | 69.72% ± 12.30% | 0.4886 |   0.5262  | 0.5096 | 0.7140 | 0.7752 |
> > >
> > > ---
> > >
> > > From the results, we can find that **not all AED methods** can achieve a better result when policy performance decreases because **they are also affected** by the dropping of demonstration quality. Notably, our proposed **PrObe is the only method** that achieves better results for all cases - *kindly compare the first and second rows and the first and third rows for each policy*.
> > >
> > > Regarding our previous forecasted result, it can be observed in *the case of NaiveDC policy (all optimal vs. all sub-optimal)* because there is no significant difference in the policy performances.
> > >
> > > As expected, the LSTMED and the TaskEmbED obtained similar AUPRC scores. SVDDED baseline suffered in this situation since it averages every demonstration frame (including those where errors occurred). DCTED and our PrObe get a slightly better result.
> > >
> > > In conclusion, the AED methods without accessing the demonstrations or intermediate process of demonstrations will produce a similar AUPRC score. Besides, the AED methods that have filtered useful information from demonstrations (e.g., DCTED and PrObe) can achieve a better result.
> > >
> > > ---
> > >
> > > Thank you again for this interesting question.
> > >
> > > Besides, feel free to let us know if there is any other experiment the reviewer would like to check.

---

> > > > ### Comment · Reviewer_upjt · 2023-11-16
> > > >
> > > > Thank you for the new experimental results! This is insightful and addresses my concerns!

---

> > > > > ### Author Response · Authors · 2023-11-16
> > > > > **Response to Reviewer upjt's comment**
> > > > >
> > > > > Dear Reviewer upjt,
> > > > >
> > > > > Thank you very much for your feedback!
> > > > >
> > > > > We are glad our new experimental result and explanation addressed your concerns. Please consider raising your score if all your concerns are resolved.
> > > > >
> > > > > Or, please let us know about any concerns you still have or experiments you are interested in; we are devoted to clarifying or checking it for you.
> > > > >
> > > > > Thank you again for providing insightful comments and valuable references.

---

### Official Review · Reviewer_BbBE · 2023-10-30

**Soundness:** 3 good
**Presentation:** 3 good
**Contribution:** 3 good
**Rating:** 5
**Confidence:** 3

**Summary:**

This work formulates the adaptable error detection (AED) problem for monitoring few-shot imitation (FSI) policy behaviors. The main goal is to have a robust system that can notify operators when FSI policies are inconsistent with the intent of demonstrations. To address the three challenges while achieving the main goal, this work proposes Pattern Observer (PrObe) parse the state patterns in policy feature representations. Experiments demonstrated that PrObe outperforms strong baselines in seven complex multi-stage FSI tasks.

**Strengths:**

1.	This work first emphasizes the importance of the adaptable error detection (AED) problem in few-shot imitation. Then, it indicates the previous error detection methods are infeasible in the three challenges of AED: (1) working in novel environments, (2) no impulse signal when errors occur, and (3) it requires online detection.

2.	Pattern Observer (PrObe) is then proposed to address the three challenges in AED. They first compute the policy feature embeddings for the successful and failed rollout frames. Then, PrObe uses an extractor to parse patterns in the policy features. Next, PrObe leverages existing temporal information by computing the pattern flow over time. Finally, PrObe distinguishes the feature source by comparing the fusion of pattern flow and transformed task embeddings.

3.	This work designed challenging multi-stage FSI tasks to validate PrObe’s effectiveness. The experiments demonstrated that PrObe outperforms baselines significantly.

**Weaknesses:**

1.	As discussed in related work, adaptable error detection (AED) is more challenging than few-shot anomaly detection (FSAD), and previous error detection methods are infeasible for AED. After reading the introduction, I understand that the main goal of AED is to monitor FSI policies and report their behavior errors. PrObe is proposed to address the challenges of AED. However, sections 4 and 5 are relatively not well connected to each other. If Figure 2 describes the whole framework of this paper, which part does PrObe contribute to? In AED training of AED inference? What is the relationship between equations (1) (2) and (3)?

**Questions:**

1.	Regarding the rollout augmentation (second paragraph of section 5.1), the authors iterate each frame from sampled agent rollouts and randomly apply the following operations: keep, drop, swap, and copy, with probability 0.3, 0.3, 0.2, and 0.2.  Is there any specific reason for choosing the four operations with the mentioned probabilities?

2.	If Figure 2 describes the whole framework of this paper, which part does PrObe contribute to? In AED training of AED inference? What is the relationship between equations (1) (2) and (3)?

---

> ### Author Response · Authors · 2023-11-11
>
> Dear Reviewer BbBE,
>
> We sincerely appreciate your insightful comments. We now address the questions and issues you raised.
>
> *The issues listed in the weaknesses and questions are denoted as Wx and Qx, respectively, where x is the issue's index.*
>
> ---
>
> **Which parts of the framework in Figure 2 do our PrObe contribute to? (W1, Q2)**
>
> Thanks for the question.
>
> Our PrObe is an AED method (i.e., adaptable error detector, denoted as $\phi$ in Figure 2), which involves the phases of AED training and AED inference.
>
> The framework of our AED problem contains three phases:
> - **Policy training phase:** The few-shot imitation policy $\pi$  first trains on the base environments.
> - **AED training phase:** The AED methods also train on the base environments. In this phase, some methods, e.g., our PrObe, might leverage the knowledge from the trained policy.
> - **AED inference phase:** The trained policy $\pi$ performs the FSI tasks in novel environments. AED methods aim to monitor the policy and report timely once the policy behavior error occurs, i.e., the policy rollout is inconsistent with demonstrations provided by the expert in the novel environments.
>
> Thus, all AED methods involve the last two phases of Figure 2; however, our PrObe can tackle the challenging AED problem with superior performance. We will make this more clear in the revised version.
>
> Besides, please note that the FSI tasks are semantically similar in the base and novel environments (e.g., both are pick&place tasks). However, the interactable objects and backgrounds are disjoint between them.
>
> **A practical scenario of our AED problem:**
>
> An intelligent robot (containing policy and AED method) is trained in the factory with daily life tasks. Then, it is sold to the customer's house to perform those tasks. The customer will provide few demonstrations for the robot but may not stay around the robot for the whole time. The robot itself should be able to recognize if it performs the task correctly or not. Also, it should terminate the policy and wait for expert's help or apply the recovery policy (in our appendix) when it detects a behavior mismatch.
>
> ---
>
> **What's the relationship between Eq. 1, 2 and 3? (W1, Q2)**
>
> Thanks for the question.
>
> Eq. 1 and 2 are the formal representations of how the AED method predicts the error status and how they will be evaluated.
>
> - Eq. 1: The AED method $\phi$ takes rollout history (observations) $h$ and few demonstrations $\mathcal{D}^{n}$ as inputs and then predicts the error status $\hat{y}$.
> - Eq. 2: The AED method is evaluated by its prediction $\hat{y}$ and ground-truth $y$ across all novel environments.
>
> Eq. 3 is a new objective we proposed to train the PrObe, which is a temporal-aware variant of triplet loss. The complete objectives of PrObe can be found in Appendix Section B.
>
> ---
>
> **Why do we choose these operations and probabilities in our designed rollout augmentation (RA)? (Q1)**
>
> Good question.
>
> As mentioned in the main paper's Section 5.1, the four operations (keep, drop, copy, and swap) can inject **different behaviors** into the pre-collected rollouts during the AED training phase.
>
> - **Dropping** consecutive frames can *speed up* the movements.
> - **Copying** consecutive frames can *slow down* the movements.
> - **Swapping** consecutive frames can provide *unsmooth* movements.
>
> Moreover, the copy operation could be replaced by the interpolation operation. We have tested this replacement, but no significant performance difference is observed. Meanwhile, the interpolation operation is very time-consuming. Thus, we still apply the four operations above.
>
> Regarding determining their probabilities (0.3, 0.3, 0.2, 0.2), we observe that the policy usually *behaves faster* than the pre-collected rollouts. Thus, we would like to have a higher chance of generating speeded-up rollouts during training.
>
> Nevertheless, the probabilities can be easily adjusted to cover more diverse rollout behaviors, which may benefit the AED training. If the reviewer wants to try any combination of the probabilities, we would love to verify it.
>
> However, the AED methods' performance is not sensitive to the probability setting, as we tested before.
>
> ---
>
> Please let us know if our responses have addressed your questions and issues.
>
> Besides, feel free to point out if there is any other issue or topic you are interested in or think will make our work more convincing.
> We look forward to having active and insightful discussions with you!
>
> Thanks again for providing valuable comments.

---

> > ### Author Response · Authors · 2023-11-17
> > **Did we resolve your concerns?**
> >
> > Dear Reviewer BbBE,
> >
> > Since we only have around five days left for the author-reviewer discussion, we would like to know if our responses have resolved your questions. If so, we would greatly appreciate you considering updating your scores.
> >
> > Otherwise, please let us know the unsolved concerns or the experiment you are interested in.
> > We are devoted to clarifying or checking it for you.
> >
> > Thank you again for providing insightful comments and valuable questions.

---

### Official Review · Reviewer_Zod3 · 2023-11-01

**Soundness:** 2 fair
**Presentation:** 1 poor
**Contribution:** 2 fair
**Rating:** 5
**Confidence:** 3

**Summary:**

The paper proposes a problem called AED to monitor few-shot imitation (FSI) learning policies. It also proposes a solution for this problem called Probe that utilizes FSI policy features.

**Strengths:**

1. AED is a fairly novel problem
2. A variety of experiments across the newly created FSI tasks.

**Weaknesses:**

1. It is very hard to read the paper. The prose could use significant improvements. There are also too many abbreviations in the paper making it hard to follow the content. What does "No impulse signals in behavior errors" mean? The problem and its challenges itself are unclear.
2. The papers makes many claims without supporting evidence, e.g. "We emphasize that addressing AED is as critical as enhancing FSI policies’ performance".
3. As far as I understand, the term "vFSAD" refers to OOD detection in time-series data which includes a large quantity of work that has not been compared with or even mentioned. Example: Kaur et al. "CODiT: Conformal Out-of-Distribution Detection in Time-Series Data."
4. The related work subtitles of "Policy" and "Evaluation tasks" are not descriptive and seemingly unrelated to the actual content.
5. There is repeated, unnecessary emphasis on the importance of the task: "FSAD is an innovative research topic", "FSI is a framework worthy of attention".
6. The stated aim of the experiments is confusing -- do you aim to check if detection is useful or if detection works?

**Questions:**

Please see weaknesses

---

> ### Author Response · Authors · 2023-11-11
> **Response to Reviewer Zod3's review [1/2]**
>
> Dear Reviewer Zod3,
>
> We respectfully disagree with your assessment.
>
> We will address the raised concerns now, and look forward to having a constructive conversation with you.
>
> *The issues listed in the weaknesses and questions are denoted as Wx and Qx, respectively, where x is the issue's index.*
>
> ---
>
> **The problem and its challenges itself are unclear. What does "No impulse signals in behavior errors" mean? (W1)**
>
> We propose a new problem named adaptable error detection (AED), which aims to detect the behavior inconsistency between the few-shot imitation policy rollouts and demonstrations.
>
> The proposed AED contains three challenges, which makes it more challenging than an existing video few-shot anomaly detection problem. The second challenge, “no impulse signals in behavior errors,” means no significant difference between the consecutive frames when the error happens. We also provide two cases with detailed visual explanations in Figure 1.
>
> It is worth mentioning that all other reviewers agree **our proposed AED problem is crucial** for robot learning when considering the safety perspective.
>
> ---
>
> **Presentation Issues (W1, W5)**
>
> We are perplexed about these issues.
>
> For W1, there are only three new abbreviations in our paper, i.e., AED for adaptable error detection, PrObe for Pattern Observer, and vFSAD for video few-shot anomaly detection.
>
> All remaining abbreviations are well-known or the method names from existing literature. For example, OCC for one-class classification, OOD for out-of-distribution, or DC for demonstration-conditioned. Also, we usually write the full name of the abbreviation at the beginning of each section if it does not frequently appear.
>
> For W5, we do not find any inappropriateness when emphasizing the importance of our work since we propose a new problem in this paper. Most importantly, the two cases the reviewer mentioned are not even our proposed problem. Instead, they are the two problems related to us. We state the fact that they are relatively new and have drawn the researcher's attention recently.
>
> At last, we want to highlight that we got >= 3 presentation scores from all other reviewers. We would appreciate it if the reviewer could point out which parts of the paper are unclear so that we can further improve it.
>
> ---
>
> **The papers make many claims without supporting evidence, e.g. "We emphasize that addressing AED is as critical as enhancing FSI policies’ performance". (W2)**
>
> It is the motivation of our work, which we spent **more than 13 lines** (first two paragraphs of Introduction) to describe the situation that the few-shot imitation policies may provoke damage when applying them to real-world scenarios. Thus, we should also consider promoting their ability for safety executions rather than the performance of their own tasks (e.g., manipulation tasks).
>
> We believe this motivation is clearly presented, which all other reviewers recognize.
>
> We would appreciate it if the reviewer could point out other unsupported claims or motivations we made in the paper. We would love to explain or improve it.
>
> ---
>
> **Works related to OOD detection in time-series data should be compared. (W3)**
>
> Thank you for pointing out this question.
>
> The fundamental assumption of our work is that we are monitoring the few-shot imitation (FSI) policies, and they usually have the following settings:
>
> - **Training and testing environments are disjoint**. (base env. vs. novel env.)
> - **No exploration before inference** in novel environments.
> - **The expert only provides demonstrations** in the novel environment but may not be around the robot when it executes.
>
> Thus, the traditional methods on OOD detection in time-series data are unavailable. In our case, the error detector works in an unseen environment. And we only have few demonstrations in the environment which makes it hard to compute those p-values in the reference [1] or other information in the frequency space.
>
> Moreover, the OOD samples defined in our work is the inconsistency between the policy rollout and demonstrations, which have two time-series. The OOD samples can be only detected by comparing these two video streams rather than simply computed on any one of them.
>
> The vFSAD problem is the most related problem we can find. But our problem raises more challenges than the vFSAD problem, as we stated in the Introduction.
>
> Please let us know if there is any other paper the reviewer thinks is related to us or fits our setting and should be compared.
>
> Thank you again for suggesting the reference.
>
> ---
>
> Please find the rest of our response below.

---

> > ### Author Response · Authors · 2023-11-11
> > **Response to Reviewer Zod3's review [2/2]**
> >
> > **The related work subtitles of "Policy" and "Evaluation tasks" are not descriptive and seemingly unrelated to the actual content. (W4)**
> >
> >
> > First, the “policy” and “evaluation tasks” **are not the subtitles of our related works**.
> >
> > They are the paragraph titles under the subsection of few-shot imitation (FSI) literature. In each paragraph, we summarize how the FSI policies develop and how FSI works design evaluation tasks to verify them.
> >
> > We did not find any inappropriateness in the naming of paragraph titles.
> >
> > ---
> >
> > **The stated aim of the experiments is confusing -- do you aim to check if detection is useful or if detection works? (W6)**
> >
> > In this work, we formulate **a new problem**. We also propose **a tailored method**, PrObe, to tackle the problem, which leverages the auxiliary knowledge of policy features.
> >
> > Therefore, our experiments aim to verify whether learning in the feature space of policy benefits to tackle the problem, i.e., **our motivation and strategy are useful and supported**. Thus, we examine tremendous evaluations across different tasks and policies. We also provide detailed ablation studies and visualization results.
> >
> > Meanwhile, we would like to express **the practicality of the proposed problem**, and thus, we conduct the error correction experiment in Appendix, which incorporates our problem with the error correction strategies.
> >
> > We believe that *the aims of our experiment are clear, and the evaluations of our experiment are comprehensive*.
> >
> > ---
> >
> > Please let us know if our responses have addressed your questions and issues.
> >
> > We will appreciate it if the reviewer can re-evaluate our work and consider the assessment from other reviewers.

---

> ### Comment · Reviewer_Zod3 · 2023-11-12
> **Additional clarification on presentation issues**
>
> As the authors requested, here are just a few issues with the writing at the start of the paper that made it hard for the reviewer to understand the paper. I've provided rewritten statements or rewritten statements phrased as questions if I was not sure how to rephrase a sentence.
>
> > adjacency frames
>
> --> "adjacent frames"
>
> > Online detection (lacking global temporal information)
>
> --> do you mean "Online detection has to take place without any information about the timestamp of the particular state"?
>
> > a robot solves a set of everyday missions by giving a few demonstrations from the owner
>
> --> "a robot solves a set of everyday tasks given few demonstrations of performing these tasks". Your original sentence made it unclear if the robot provides the owner of the robot with demos or if the robot is given demos.
>
> > task misunderstanding [Figure 1]
>
> --> I am not sure what this image even represents. Is the agent doing the opposite of the expert demonstration?
>
> > ... learning in policy features also has additional knowledge related to FSI tasks
>
> --> learning policy features provides additional knowledge
>
> > (1) AED monitors the policy in novel environments whose normal states are not seen during training. Although vFSAD
> also monitors unseen videos, the anomalies they monitor are objects, which in AED are policy behaviors. (2) Unlike the anomaly in vFSAD raises a notable change in the perception field (e.g., a cyclist suddenly appearing on the sidewalk (Mahadevan et al., 2010)), no impulse signals appear when behavior errors occur. Either (2-a) minor visual difference or (2-b) task misunderstanding is hard to recognize through adjacency rollout frames. (3) AED requires online detection to terminate
> the policy before causing disasters, lacking global temporal information
>
> --> It is unclear how a problem (AED or vFSAD) "does" anything. These words are used without any explanation: "impulse signals", "global temporal information".
>
> But, here is my attempt to rephrase your paragraph: "(1) In the AED problem, unlike the vFSAD problem, novel anomalies are actions performed by a policy rather than objects that appear in a video. (2) The observed changes are subtle in the AED problem unlike vFSAD. (3) The AED problem, unlike vFSAD which detects in anomalies in videos that have already been recorded, does not have any information from the future and hence is more challenging."
>
> > unvanquishable obstacles
>
> --> challenging obstacles
>
> > ... to demonstrate the practice of the proposed AED problem
>
> --> ... to demonstrate potential next steps after the AED problem is addressed.
>
> >  the community explores the paradigm for policy learning ...
>
> --> the wider community has explored various solutions for learning from demonstrations such as ...
>
> > Notably, they either assume the agent and expert are the same or explicitly learn an action mapping
>
> --> I am not sure what this could mean.
>
> > MoMaRT forms a more challenging task using a mobile robot
>
> ---> MoMaRT tackles a more challenging mobile robot task, namely ...
>
> > demonstrations are length-variant
>
> --> demonstrations vary in length
>
> > Developing a policy to solve the FSI problem and simultaneously tackle these challenges is indispensable
>
> --> indispensable to who? Perhaps you mean "Developing a policy to solve the FSI problem and simultaneously tackle these challenges is difficult but necessary for human-robot collaboration"
>
> > After receiving the rollout history h, the feature encoder extracts the history features f_h. Meanwhile, the feature encoder also extracts the demonstration features. Then, the task-embedding network computes the task-embedding f_ζ.
>
> --> Is f_h the same as f_ζ. What is f even? A function is usually defined by a domain and a codomain before being used.
>
> > an adaptable error detector ϕ
>
> --> again, no definition for ϕ
>
> > ϕ predicts the categorical probability y of the behavior mode for the lastest state in h
>
> --> I am not sure what the behavior mode refers to. Also, "lastest state" --> "last state"
>
> > The task-embedding network is expected to handle length-variant sequences by padding frames
>
> --> The task-embedding network is expected to handle input sequences of varying length with padding to a large prefixed length.
>
> > Our main experiment examines whether the AED methods can accurately report behavior errors when seeing the policy rollouts.
>
> --> Our main experiment examines whether the method aimed at solving the AED problem can accurately report behavior errors when provided policy rollouts.
>
> I am happy to help in identifying other statements that could use improvement but believe the above few represent a good starting point.

---

> > ### Author Response · Authors · 2023-11-13
> > **Response to the reviewer's comment**
> >
> > Dear Reviewer Zod3,
> >
> > We sincerely appreciate the reviewer's detailed suggestions for improving our paper writing; we will revise those typos and phrases that may cause ambiguity.
> >
> > Meanwhile, we are glad the reviewer can correctly capture the meaning through our original sentences in most cases. Nevertheless, we would like to thank the reviewer again and try our best to use these beautiful and detailed sentences in the paper with page limits.
> >
> > ---
> >
> > Moreover, we also want to learn more from the reviewer's comments on our work.
> >
> > Our work proposes a new problem, *adaptable error detection (AED)*, for monitoring the **inconsistency** between few-shot imitation policy rollouts and demonstrations, i.e., behavior errors, to promote attention on robots safely performing tasks.
> >
> > AED problem brings three challenges, clearly stated in Figure 1:
> > 1. Detecting errors in the **unseen environment**.
> > 2. **no impulse signals** when behavior errors occur. Here, "no impulse signals" means we cannot observe the notable changes through consecutive frames, as stated in the subtitle of challenge 2 in Figure 1. We also provide an opposite case that will cause impulse signals, the anomaly in vFSAD.
> > 3. It requires **online detection** during the rollout execution, which lacks the global temporal information of the rollout. We mark this challenge at the timestamp part of Figure 1. And "future steps" indicate that the rollout is not complete.
> >
> > To the best of our knowledge, the existing methods cannot resolve our AED problem. Thus, we propose a tailored PrObe method to address it via *learning in the policy feature space*.
> >
> > Besides, we conducted detailed experiments to prove PrObe's effectiveness. By being validated on seven challenging few-shot imitation tasks across different domains and environments, our PrObe outperforms several **strong baselines with various characteristics**. We also demonstrate the **practical usage of our AED problem** in the error correction experiment.
> >
> > We are grateful that the reviewer recognized our effort in our experiment evaluations.
> >
> > Therefore, we would love to learn how to improve our work from the reviewer.
> >
> > ---
> >
> > We thank the reviewer again for being willing to discuss with us about our work.

---

> ### Comment · Reviewer_Zod3 · 2023-11-22
>
> My concerns beyond the writing have been addressed by the authors response and my (fifth!) rereading of the paper. Perhaps my initial rating was a bit harsh, and hence, I've amended my rating to a 5 (although I am leaning towards 3). The problem is novel. The FSI tasks to evaluate Probe are comprehensive. But, since I believe that the writing needs a big update to be suitable for ICLR (and that has not been done so far by the authors), I am not recommending an accept considering the significant effort required by anyone to parse the first half (before experiments) of the paper.

---

> ### Author Response · Authors · 2023-11-23
> **Response to Reviewer Zod3's comment**
>
> Dear Reviewer Zod3,
>
> Thank you for the valuable feedback and your time in reviewing our paper.
>
> First, we sincerely appreciate your updated score and positive feedback on our work (related to the problem setting, method, and evaluation).
>
> Besides, we agree that improving paper writing can strengthen our work and make it easier for all readers to follow.
> Thus, we are thankful for your writing suggestions in our earlier conversations.
>
> To reflect our determination to continuously improve our paper writing, we have considered most of your suggestions and updated our paper. Please check the updated PDF (*modified parts are marked as blue*). We keep some parts as they are since we need more time to rephrase the paragraphs. But we will carefully inspect the whole paper and continually improve it.
>
> Hence, if the updated PDF eases some doubts about paper writing and you recognize our problem, method, and evaluation, Please consider further updating your evaluation.
>
> We will deeply appreciate it and try our best to address this remaining concern.
>
> Thank you again for your valuable time to review our work and your active participation in the discussion.

---

### Author Response · Authors · 2023-11-22
**Discussion period is coming to an end**

Dear all reviewers,

Thank you for providing feedbacks during the discussion.

As the discussion deadline is approaching, we would like to know if our response has addressed your concerns.

During the discussion, we conduct **one more experiment on the demonstration qualities**, which is suggested by *reviewer upjt*.
We are glad that our new results and explanations addressed the reviewer upjt's concerns.

Besides, we also provided **detailed explanations** on the questions raised by other reviewers. We would love to hear the reviewers' feedback and learn more.
Please feel free to continue our conversations if you have any further questions.

If our response has cleared your concerns, we would greatly appreciate it if you could consider updating your score.

Thank you again.

---

### Meta-Review · Area_Chair_A5UJ · 2023-12-08

**Metareview:**

This paper defines a problem of adaptatable error detection (AED): detecting errors in imitation-learned policies for few-shot settings so that the policy roll-out can be terminated. The problem is challenging due to failures/successes often being recognizable based on a combination of the observations and policy characteristics, and recognition needed in online, novel settings. This seems to be an interesting and important problem for practical applications in robotics. The paper develops Pattern Observer (PrObe) to identify patterns corresponding to normal/error states. A fairly extensive evaluation is performed using robotic manipulations showing that the PrObe approach outperforms other anomaly detection methods.

The paper makes good contributions, but these need to be more clearly conveyed to the reviewers to be convincing. Specifically, the notion of policy-dependence/independence seems to perhaps be crucial for understanding the benefits of the approach, but it isn't clearly articulated in the paper. Instead, "no impulse signals" is described almost as an assumption rather than a general result of the characteristics of the problem. Claims that are made without sufficient evidence (e.g., "addressing AED is as critical as enhancing FSI policies’ performance") are also not convincing. Based on this, I think the paper is borderline, but side with the majority of reviewers to recommend against acceptance.

**Justification For Why Not Higher Score:**

The paper makes good contributions that need to be more clearly conveyed (with better supporting arguments) to convince the reviewers.

**Justification For Why Not Lower Score:**

N/A

---

### Decision · Program_Chairs · 2024-01-16

Reject